



**Interactions between biogeochemical and management factors explain soil**
**organic carbon in Pyrenean grasslands**
**Authors**
**Antonio Rodríguez[1,2], Rosa Maria Canals[3], Josefina Plaixats[4], Elena**
**Albanell[4], Haifa Debouk[1,2], Jordi Garcia-Pausas[5], Leticia San Emeterio[6,7], Juan**
**José Jimenez[8], M.-Teresa Sebastià[1,2]**
**Afiliations**
1: Group GAMES and Department of Horticulture, Botany and Landscaping, School of
Agrifood and Forestry Science and Engineering, University of Lleida, Lleida, Spain
2: Laboratory of Functional Ecology and Global Change (ECOFUN), Forest Sciences
Centre of Catalonia (CTFC), Solsona, Spain
3: Grupo Ecología y Medio Ambiente and ISFood Institute, Universidad Pública de
Navarra, Campus Arrosadia, Pamplona, Spain.
4: Group of Ruminant Research (G2R), Department of Animal and Food Sciences,
Universitat Autònoma de Barcelona, 08193, Bellaterra, Spain
5: Forest Sciences Centre of Catalonia (CTFC), Solsona, Spain
6: Research Institute on Innovation & Sustainable Development in Food Chain
(ISFOOD), Universidad Pública de Navarra,31006 Pamplona, Spain.
7: Departamento de Agronomía, Biotecnología y Alimentación, Universidad Pública de
Navarra, 31006 Pamplona, Spain.
8: ARAID/IPE-CSIC, 22700, Huesca, Spain
**ORCID**
Antonio Rodríguez http://orcid.org/0000-0002-0536-9902





Elena Albanell https://orcid.org/0000-0002-6158-7736
Jordi Garcia-Pausas https://orcid.org/0000-0003-2727-3167
Leticia San Emeterio http://orcid.org/ 0000-0002-8063-0402
M.-Teresa Sebastià http://orcid.org/0000-0002-9017-3575

**Author contributions**

Antonio Rodríguez designed the statistical procedure, carried out the statistical analyses
and wrote the original draft.
Rosa M Canals was responsible from field monitoring, lab analyses and acquisition of
information for the data base implementation in the Western Pyrenees (Navarra). She
has reviewed the draft.
Elena Albanell designed NIRS study and reviewed the draft.
Haifa Debouk sampled and processed some of the data in the PASTUS Database and
reviewed the draft.
Jordi García-Pausas processed some of the data in the PASTUS Database and
reviewed the draft.
Josefina Plaixats carried out the chemical analisys of herbage samples for NIR
calibration and validation equations and reviewed  the original draft.
Leticia San Emeterio designed methodology and data collection, performed soil and
vegetation sampling. She has reviewed the draft.
Juan José Jiménez collaborated in the fieldwork and reviewed the draft.
M.-Teresa Sebastià contributed to the conception, design and development of the
PASTUS database. In addition, she ensured funding and coordinated the projects whose





data are included in PASTUS. Finally, she contributed to initial modelling, supervised the
development of the paper, and read and  reviewed the drafts.



## Abstract

Grasslands are one of the major sinks of terrestrial soil organic carbon (SOC).
Understanding how environmental and management factors drive SOC is challenging
because they are scale-dependent, with large scale drivers affecting SOC both directly
and through drivers working at detailed spatial scales. Here we addressed how regional,
landscape and grazing management, soil properties and nutrients and herbage quality
factors affect SOC in mountain grasslands in the Pyrenees. Taking advantage of the high
variety of environmental heterogeneity in the Pyrenees, we fit a set of models with
explicative purposes using data that comprise a wide range of environmental and
management conditions. We found that temperature seasonality (TSIS) was the most
important geophysical driver of SOC in our study. TSIS was positively related to SOC
but only under certain local conditions: exposed hillsides, steep slopes and relatively
highly grazed areas. High TSIS conditions probably are more favourable for plant
biomass production, but landscape and grazing management factors buffer the
accumulation of this biomass into SOC. Concerning biochemical SOC predictors, we
obtained some surprising, interactive effects between grazer type, soil nutrients and
herbage quality. Soil N was a crucial factor modulating effects of livestock species and
neutral detergent fibre content of plant biomass and herbage recalcitrance effects varied
depending on grazer species. These results highlight the gaps in the knowledge about
SOC drivers in grassland under different environmental and management conditions,
and they may serve to generate testable hypothesis in latter studies directed to climate
change mitigation policies.

## Keywords

SOC, semi-natural grasslands, grazing management, climate change, soil nutrients

## Introduction





Soil organic carbon (SOC) plays key roles in the terrestrial ecosystems (Lal,
2004a). SOC enhances soil and water quality and biomass productivity, and has
an important role in relation to climate change (Lal, 2004b). Although grasslands
have small aboveground biomass compared to other ecosystems, their SOC
stocks can be comparable to those in forest ecosystems (Berninger et al., 2015).
This is due to their high root biomass and residues, which are a substantial
carbon source and can contribute to water retention in soil. This creates
favourable conditions for the accumulation of organic matter (Von Haden and
Dornbush, 2014; Yang et al., 2018). These attributes, together with the high
extent of grassland global cover, make grasslands store around 34% of the
terrestrial carbon, mostly in their soils (White et al., 2000).
SOC accumulation results from a complex equilibrium between primary
production and organic matter decomposition which depends on multiple
environmental factors such as climate, soil texture and nutrients or land
management (Jenny, 1941; Schlesinger, 1977). Understanding how these
environmental factors drive SOC is challenging because they are scale-
dependent and are disposed on a hierarchy of controls over SOC, so large scale
drivers affect also those working at fine spatial scales (Fig. 1; Manning et al.,
2015). Climate is known to be the main SOC driver at broad (global and regional)
scales; mean annual precipitation (MAP) and mean temperature (MAT) being
the most frequent climate indicators (Wiesmeier et al., 2019). However, climate
seasonality variables are be commonly neglected drivers affecting SOC in broad-
scale models, in spite of being some important factors for plant primary
production or enzymatic activity of soil microorganisms (Fernández-Alonso et al.,
2018; Garcia-Pausas et al., 2007; Puissant et al., 2018). Other regional and



landscape factors like bedrock or topography are also considered to be at the top
of the hierarchy because they influence multiple geophysical and biochemical
factors affecting SOC, including soil texture or water flow paths (Gray et al., 2015;
Hobley et al., 2015). Next in the hierarchy after regional and landscape factors,
are several soil geophysical properties, like pH and texture, which are controlled
by climate, bedrock, and which affect SOC through both plant primary production
and microbial activity and the capacity to stabilise the SOC (Deng et al., 2016;
Xu et al., 2016a). Soil macro and micronutrients are in the next level of the
hierarchy, as their abundance is determined by multiple factors, including climate,
soil pH, water content or clay content (Hook and Burke, 2000; de Vries et al.,
2012). They play an essential role in primary production and herbage quality, and
act as resources for microbes to mineralise SOC (Aerts and Chapin, 1999;
Vitousek and Howarth, 1991). However, these variables are commonly omitted
in the broad-scale SOC studies, especially if those focus on predictive models
instead of explicative ones (Gray et al., 2015; Manning et al., 2015; Zhang et al.,
2018). This kind of variables are less frequently available and more difficult to
measure than the other indicators used for SOC modelling (Manning et al., 2015).
Moreover, the use of soil nutrients as SOC predictors in linear models can be
challenging, as they are often so linked to SOC dynamics that their effect can
mask the effect of other predictors at higher levels (Bing et al., 2016; Cleveland
and Liptzin, 2007; Tipping et al., 2016). Vegetation represents another group of
SOC predictors, affected by climate, topography and soil properties and nutrients
(Fernández-Martínez et al., 2014; de Vries et al., 2012; Zhu et al., 2019). Plant
biomass is the main input of organic carbon into the soil (Shipley and Parent,
1991). However, plant litter quality can determine decomposition rates and





patterns, and hence soil carbon sequestration (Ottoy et al., 2017; Yan et al., 2018,

126  2019).

Apart from these factors, management effects on grassland SOC is a noteworthy
issue since they are poorly understood (Wiesmeier et al., 2019). It is known that
herbivores can affect SOC through different paths, such as regulating the quantity
and quality of organic matter returned to soil (Bardgett and Wardle, 2003), or
affecting soil respiration and nutrients by animal trampling or soil microbiota
alteration (Lu et al., 2017). However, most of the studies investigating grazing
effects on SOC focus on grazing intensity, in spite of evidence pointing to a
greater role of grazer species in determining vegetation and SOC (Chang et al.,
2018; Sebastia et al., 2008). Moreover, several studies describing interactions of
grazing with other SOC predictors at diverse scales have been published (Abdalla
et al., 2018; Eze et al., 2018; Lu et al., 2015, 2017; Zhou et al., 2017). Hence,
grazing management on grasslands may be considered a unique SOC driver,
because it has effects at multiple levels of the driver hierarchy (Fig. 1).
In this study, our goal was to identify the main drivers of SOC stocks in semi-
natural grasslands of the Pyrenees, asses the interactions between them and
describe their relative importance. Mountain grasslands comprise a wide range
of climatic, topographic, management and edaphic conditions that make carbon
stocks highly variable (Garcia-Pausas et al., 2007, 2017). For this reason data
analysed here comprise a wide range of environmental conditions, comparable
to studies on SOC developed at continental or even worldwide scales (Table 1).
Additionally, we consider an exceptionally broad compilation of predictors (Table
1). In particular, the specific questions of this study are  1) how are the effects of
the geophysical, widely used predictors located at the top of the hierarchy of



controls on SOC? 2) how are the effects of the biochemical, unfrequently used
(soil nutrient and herbage), predictors on SOC? 3) Can grazing management
regulate the effects of other SOC drivers located at different levels of the
hierarchy of controls?
**Material & methods**
**2.1 Location and sampling design**
The set of data used in this study has been extracted from the PASTUS Database
(http://ecofun.ctfc.cat/?p=3538), which was compiled by the Laboratory of Functional
Ecology and Global Change (ECOFUN) of the Forest Sciences Centre of Catalonia
(CTFC) and the University of Lleida (UdL). We sourced a wealth of data of 128 grassland
patches distributed across the Pyrenees (Fig.S1), and including topographical,
climatological, soil, herbage and management variables. The sampled area
encompasses a wide variety of temperate and cold-temperate climates, with different
precipitation conditions, depending on altitude and geographical location from
Mediterranean to Continental and Boreo-Alpine (de Lamo & Sebastià, 2006; Rodríguez
et al., 2018; Table 1).
Sampling in the PASTUS database was designed according to a stratified random
scheme, where samples were selected at random within strata. This process was done
using the software ArcMap 10 (ESRI, Redlands, CA, USA). The basis for randomization
was the map of habitats of Catalonia 1:50000 (Carreras and Diego, 2006) for the Eastern
and Central sectors, the map of habitats of Madres-Coronat 1:10000 (Penin, 1997) for
the North-Eastern sector and the land use map of Navarra 1:25000 (Gobierno de
Navarra, 2003) for the Western sectors. Four variables were initially considered for
sampling stratification within each sector: altitude (< 1800 m; 1800-2300 m; > 2300 m),
slope (0-20º; 20-30º; > 30º), macrotopography (mountain top/southern-facing slope;
valley bottom/northern-facing slope) and grazing management (sheep grazing; cattle

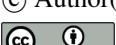



grazing; mixed grazing). Accordingly, we determined a set of homogeneous grassland
patches by crossing the stratification variable layers. Grassland patches were then listed
by type and arranged within each list randomly to determine sampling priority. At least
one to two replicates of each patch type were sampled.
In each sampled grassland patch, a 10 x 10 $m^2$ plot was systematically placed in the
middle of each homogeneous grassland patch, including a particular plant community.
Soils and vegetation were sampled inside this 100 $m^2$ plot, and environmental variables
assessed (see Rodríguez et al., (2018) for additional details about sampling design).
Local variables were assessed inside the 100 $m^2$ plots. Aboveground biomass was
estimated from herbage cut at ground level in four 50 x 50 $cm^2$ quadrats placed in a 2 x
2 $m^2$ subplot inside the 100 $m^2$ plot. Herbage from two of the four quadrats were dried
and sent to the laboratory for duplicated chemico-bromatological analysis. In addition, in
each quadrat, a 20-cm depth soil core was extracted with a 5 x 5 cm probe after herbage
was removed. The soil sample in the probe was separated into two soil layers: 0-10 and
10-20 cm.
**2.2 Regional and landscape environmental drivers**
In order to investigate the relationship between soil organic carbon (SOC) and potential
environmental drivers, 29 independent environmental variables were initially considered
(Table S1). These variables were grouped into five sets: Regional, landscape: livestock
management, soil nutrient stocks and herbage variables.
Regional variables included climate variables and bedrock. Climate variables were
determined from Worldclim 2.0 (Fick and Hijmans, 2017). We selected Mean Annual
Temperature (MAT), Mean Annual Precipitation (MAP) and Mean Summer Precipitation
(MSP). The difference between mean annual and mean summer temperature emerged
as a relevant explanatory factor of soil organic carbon stocks during previous modelling
efforts by one of the co-authors (MTS). Latter attempts to improve models by substituting



this variable by other temperature indices in climatic databases (Fick and Hijmans, 2017)
showed that, for the PASTUS database, this variable provided higher explanatory power
than other temperature seasonality indices. Thus, we decided to keep it and here we
name it Temperature Seasonality Index of Sebastià (TSIS from now on).
Bedrock type was determined in the field and confirmed by the geographical maps
mentioned above. Bedrock was categorized into three categories: basic (marls and
calcareous rocks), acidic (mostly sandstones and slates) and heterogeneous.
Landscape variables included topography and soil type variables. Topography variables
included Slope, Aspect, Macrotopography and Microtopography. Slope and Aspect were
determined in the field by clinometer and compass respectively. Macrotopography and
Microtopography were determined visually in the field. Macrotopography differenciated
exposed from protected positions. Mountain top and south-facing slopes were identified
as exposed positions and valley bottoms and north-facing slopes as protected positions.
Microtopography considered three positions: convexities, concavities and smooth areas.
Soil type variables are described in section 2.4.
**2.3 Livestock management variables**
Regarding livestock management variables, detailed surveys were carried out among
farmers, shepherds and land managers. Two management variables were considered:
Grazing intensity and Grazer type. Grazing intensity was determined estimating livestock
stocking rates measured as livestock units ha$^{-1}$ (LU ha$^{-1}$), and treated as a semi-
quantitative variable with three categories (Sebastià et al. 2008): low (1; lower than 0.2
LU ha$^{-1}$), medium (2; between 0.2-0.4 LU ha$^{-1}$) and high (3; above 0.4 LU ha$^{-1}$). Grazer
type was categorized into three main types: sheep grazing, cattle grazing and mixed
grazing. Mixed grazing included associations comprising small and big livestock species,
mainly sheep and cattle, and more rarely horses. Sheep flocks always included some
goats.





**2.4 Soil sampling and physicochemical analysis**

Soil samples were air-dried and weighted. Each sample was sieved to 2 mm to separate stones and gravels from the fine earth fraction; the fine fraction was sent to the laboratory for physicochemical analysis. Standard physicochemical soil analyses were performed in the upper 0-10 cm soil layer of all grasslands. Some analyses were also performed on samples from the 10-20 cm soil layer, including: soil organic carbon, total nitrogen. For those variables, we later calculated values for the whole top 20 cm soil layer.

All soil physicochemical analyses were performed on the fine earth, according to standard soil analysis methods. Textural classes were determined by the Bouyoucos method (Bouyoucos, 1936). Soil pH (measured in water), total organic carbon (TOC) total nitrogen (TN), Calcium content (Ca), Extractable phosphorus (P), magnesium (Mg) and potassium (K) were measured on air dried samples (Schöning et al., 2013; Solly et al., 2014). Soil carbonates were determined using the Bernard calcimeter. Total carbon and nitrogen (N) contents of the fine earth was determined by elemental auto-analyser. The organic C fraction was determined by subtracting inorganic C in the carbonates from the total C. Soil organic carbon (SOC) stocks in the upper 20 cm soil layer were then estimated taking into account the organic C concentration in the sample and its bulk density, and subtracting the coarse particle (> 2 mm) content, following García-Pausas et al. (2007). Available phosphorus (P) was extracted by the Olsen method (Olsen, 1954) Ca, Mg and K were extracted by ammonium acetate (Simard, 1993) and measured by flame Atomic absorption Spectroscopy (AAS) (David, 1960)).

**2.5 Herbage chemical and bromatological analysis**

A total of two hundred samples were chemical and bromatological analysed by NIRS (near infrared reflectance spectroscopy). All four herbage samples of each plot were oven-dried at 60ºC to constant weight. Two of the samples was sent to the laboratory. Dried samples were ground to pass a 1 mm stainless steel screen (Cyclotec 1093



Sample mill, Tecator, Sweden) and stored at 4ºC until it was needed for use. To develop
NIRS equations (see below) subsamples were analysed in duplicate. Procedures
described by AOAC were used to determine dry matter (DM) and ash content or mineral
matter (MM). Crude protein (CP) was determined by the Kjeldhal procedure (N x 6.25)
using a Kjeltec Auto 1030 Analyser (Tecator, Sweden). Samples were analysed
sequentially for neutral detergent fibre (NDF), acid detergent fibre (ADF) and acid
detergent lignin (ADL) in accordance with the method described Van Soest et al. (1991)
using the Ankom 200 Fibre Analyser incubator (Ankom, USA). The fibre analysis were
determined on an ash-free basis and without alpha amylase. We calculated two
additional herbage quality indexes often used in the bibliography: NDF/CP and ADL/HN
(Stockmann et al., 2013). For each subsample the C and N content were determined by
the Dumas dry combustion method,  using an Elemental Analyzer EA1108 (Carlo Erba,
Milan, Italy).
**2.6 NIRS analysis**
NIRS data were recorded from 1,100 to 2,500 nm using a FOSS NIRSystem 5000
scanning monochromator (Hillerød, Denmark). Separate calibration equations
were generated for grassland samples. Reflectance (R) data were collected in
duplicate every 2 nm. A WinISI III (v. 1.6) software (FOSS, Denmark) was
employed for spectra data analysis and development of chemometric models.
Prior to calibration, log 1/R spectra were corrected for the effects of scatter using
the standard normal variate (SNV), detrend (DT) and multiple scatter correction
(MSC) and transformed into first or second derivative using different gap size
(nm) and smoothing interval. For each sample, the mean of the spectra from the
two lectures were used. Modified partial least square (MPLS) was the regression
method used for calibration development and cross validation was undertaken
using the standard methodology in the NIRS software program. The performance





of the model was determined by the following statistical tools: standard error of
calibration (SEC), standard error of cross validation (SECV); coefficient of
determination for calibration ($R^2$) and cross validation ($r_{cv}^2$); the ratio of
performance to deviation (RPD) described as the ratio of standard deviation for
the validation samples to the standard error of cross validation (RPD=SD/SECV)
should ideally be at least three; and the range error ratio (RER=Range/SECV)
described as the ratio of the range in the reference data to the SECV should be
at least 10 (Williams and Sobering, 1996; Williams et al., 2014).
**2.7 Statistical analyses**
We applied two different modelling procedures, Boosted Regression Trees (BTR) and
General Linear Models (GLM). All the statistical analyses were performed with the
software R ver. 3.4.3 (R Core Team, 2017), at 95% significance level when appropriate.
*Boosted regression trees global model*
We fitted a model with BRT to identify the most important variables affecting SOC. BRT
uses two algorithms: regression trees are from the classification and regression tree
(decision tree) group of models, and boosting builds and combines a collection of models
(Elith et al., 2008). We chose this method because BRT can handle multiple variables
better than other techniques as GLM, and can detect automatically curvilinear
relationships and interactions, ignoring non-informative ones. We used the gbm and
dismo packages (Greenwell et al., 2019; Hijmans et al., 2017), which provide several
functions to fit these models.
First, we fitted a model with all the predictors (Table S1), configured with 15 folds, a
Gaussian distribution of the error, a tree complexity of 5, a learning.rate of 0.005, a
bag.fraction of 0.666, and 5 minimum observations by node. Secondly, we reduced the
number of predictors by the method described in Elith et al., (2008). We estimated the





change in the model´s predictive deviance dropping one by one each predictor
(supporting information), and re-fitted the model with the set of variables which actually
improved model performance Fig. S2). We checked the relative importance of the
predictors and the shape and size of the effects by partial effect plots.
*General linear models*
We designed and executed a modelling procedure based on general linear models
(Legendre and Legendre, 1998) and a hierarchy of controls over function (Diaz et al.,
2007; de Vries et al., 2012). We log-transformed SOC using natural logarithm to prevent
a breach of the normality assumption by the residuals of the models. We built two models
(Fig. S4), one model only based on geophysical predictors and grazing management
(Geophysical Model), and another model by adding to the former the biochemical
predictors: soil nutrients and herbage quality predictors (Combined Model). We
considered that the geophysical factors that potentially affect SOC were regional and
landscape (topography and soil type predictors), as they have been widely used in
previous studies to model and predict SOC from landscape to continental scales
(Manning et al., 2015; Wiesmeier et al., 2019). In addition to soil nutrients and herbage
variables, we included again the livestock management variables in the Combined Model
and looked for interactions involving these variables and biochemical predictors of SOC.
For model building (Fig. S4A), we added predictor groups following a sequential order.
For fitting the geophysical model, we started adding regional, landscape and grazing
management predictors, and subsequently included soil properties. Afterwards, we
sequentially included soil nutrients and herbage predictors to obtain the Full Model. We
added Management variables from the beginning of the modelling process and re-
included the discarded ones in each step to guarantee the detection of interactions
between Management variables and the rest of the predictors. Each time we added a
set of predictors, we first considered their main effects and some quadratic terms which



were found by preliminary analyses with the scatterplot.matrix function in the R package
car (Fox et al., 2018); afterwards we included possible level 2 interactions between all
the selected predictors.
At every step we selected several candidate terms by a semi-automatic procedure (Fig.
S4C) using a genetic algorithm included in the R package glmulti (Calcagno, 2015). We
used SOC as response variable at the first step, and the residuals of the previous model
in the remaining steps (Fig. S4B). This semi-automatic process began by obtaining a
best subset of models using the corrected Akaike information criterion (AICc),
appropriate when n/k is less than 40, being the sample size and k the number of
parameters in the most complex model (Symonds and Moussalli, 2011). We selected the
best model and its equivalents obtained by the genetic algorithm, which were those
within 2 Akaike information criterion-corrected (ΔAICc) values of the best model, as a
ΔAICc < 2 indicates that the candidate model is almost as good as the best model
(Burnham and Anderson, 2002).
For this set of models, we built averaged models using the MUMIn package (Barton,
2015). We calculated partial standardized coefficients, obtained by multiplying the
unstandardized coefficient in the model by the partial standard deviation of the variable,
which is a function of the standard deviation of the variable, the sample size, the number
of variables in the model and the variance inflation factor of the variable (Barton, 2015).
We selected all the variables with significant effects (alone or in interaction with each
other) in the conditional average model, which was preferred over the full average model
because we wanted to avoid excessive shrinkage effects at this moment of the modelling
procedure (Grueber et al., 2011).
Then, we added these terms to the consolidated model, and made a selection through a
backward forward procedure. We used several methods to compare and determine the
final model, including the AICc, the adjusted determination coefficient $R^2$ ($R_{adj}^2$) and
model comparison techniques with the "anova()" function in R, using Chi-square tests to





test whether the reduction in the residual sum of squares was statistically significant.
Once we had the final model we assessed the significance of each term by removing it
and performing an F test. For estimating the significance of the main effects we also
removed the interaction terms in which they were involved, to avoid transferring the
effects of the main terms to the interaction terms (de Vries et al., 2012). We estimated
the variance explained by the models through the adjusted determination coefficient $R^2$
($R_{adj}^2$).
Finally, we estimated the importance of the terms of each model by the lmg method in
the relaimpo package (Grömping, 2006), and drew partial effect plots making predictions
with the R package emmeans (Lenth et al., 2019).






**Results**
SOC stocks of the upper 20 cm layer ranged between 2.6 and 23 kg m$^{-2}$, with a
median and a mean value of 9.1 and 9.6 kg m$^{-2}$ respectively. Minimum, maximum,
median and mean values of the continuous predictors are shown in Table S2.
*Relative importance of SOC predictors*
The final BRT global model achieved a good goodness of fit, with a cross-
validated correlation value of 0.52% and an explained deviance of 88.31%. The
most important variables explaining SOC stocks (Fig. 2) were soil N (18.3 %), soil
C/N (14.4%) and Clay (13 %) although other variables such as Aboveground
biomass (7.3%), ADL (6.4%) or Silt (6.1%) were also relevant for explaining SOC
storage. Two of the most important variables in the BRT model, Aboveground
biomass and Silt, were not selected in the linear models (Tables 2 & 3). Although
accounting for a lower importance value than the previous variables (5%), TSIS
was the most relevant selected climate predictor. This variable was also relevant
in both linear models (Fig. S5), especially in the Geophysical Model, where TSIS
was the most important variable, not only as main effect, but in interaction with
other variables (lmg: 4 – 10%). Soil nutrient and herbage variables were also
important according to the Combined linear model (Fig. S6), but in this case we
identified that many of these effects occurred in interaction between these two
predictors with grazer type.
Geophysical effects on SOC stocks
The Geophysical Model (Table 2) explained 34 % of the total variance ($R^2_{Adj}$).
Overall, SOC stocks increased with TSIS under certain conditions: exposed




Biogeosciences Open Access
Discussions
EGU

hillsides, high slopes and low stocking rates (Fig. 3A, 3B & 3D). On the other
hand, Clay had a positive relationship with SOC under low MAP values (Fig. 3C),
which turned into negative at high MAP values (Fig. 6C).
Soil nutrient and herbage effects on SOC
Adding nutrient and herbage predictors in the previous geophysical model to build
the Combined model (Table 3) increased the total variance ($R^2_{Adj}$) up to 61%.
Macrotopography and Clay effects described by the Geophysical model were
removed by the new model terms included. SOC increased with C/N (Fig 4A).
Soil nitrogen modulated the effects of livestock type and NDF on SOC. Cattle
grazed grasslands stored more SOC than mixed and sheep grazed grasslands
under low soil N conditions, whereas the opposite occurred at high soil N levels
(Fig. 3B). NDF had negative effects on SOC at high soil N values but had no
effect under low soil N levels Fig. 4C). Finally, herbage ADL/NH had positive
effects on SOC under mixed and sheep grazing regimes, but there was no effect
under cattle management (Fig. 4D).
**Discussion**
Regional, landscape, management, soil and herbage factors drove SOC stocks
in grasslands of the Pyrenees with multiple interactions. The BRT model identified
soil N and C/N, texture and herbage variables as the most important predictor
groups (Fig. 2), TSIS being the most important climate variable. Both linear
models followed a hierarchy of controls over function approach to ensure a
unique effect of each driver on SOC. Hence, some variables selected in the BRT
model, like aboveground biomass, silt or soil K were not included in these models
(Tables 2 & 3). The geophysical model showed how some climate variables (TSIS



and MAP) interacted with landscape (macrotopography and slope), soil clay
content and grazing intensity (Fig. 3). Whereas, the Combined Model provided
information on how herbage quality effects on SOC (NDF and ADL/NH) varied
depending on soil N and grazing species, and on how grazer species had
different effects depending on soil N content (Fig. 4).

*Considerations about the modelling procedure*
As a regression tree machine learning technique, the BTR model identified a set
of SOC predictors (Fig. 2) avoiding some of the linear model disadvantages, like
guarding against the elimination of good predictors correlated to others or
automatically modelling non-linear effects (Cutler et al., 2007; Elith et al., 2008).
Thus, the BRT model included some SOC predictors, like a positive logarithmic-
like effect of aboveground biomass or soil K on SOC (Fig. S7), which could be
masked by the effects of other variables in our linear models (Yang et al., 2009).
However, most of the variables selected and their effects were generally
consistent with those explained by the linear models (Fig. 3, 4, S7).
Consequently, we preferred  to focus on the results from the linear models
because our approximation allowed us to build models under a hierarchy of
controls over function hypothesis (Manning et al., 2015). Hence, although we
could not establish the causal links between SOC predictors (Grace, 2006; Grace
and Bollen, 2005), we guaranteed that geophysical drivers included in the first
model were not the single common cause of variation of both biotic factors
included in the second model and SOC (de Vries et al., 2012). In that case, soil
nutrient and herbage quality predictors could not be added to the model as





significant terms, as was the case with aboveground biomass. In addition, our
modelling approach allowed us to select biologically meaningful interactions
(Manning et al., 2015; de Vries et al., 2012), which cannot be done with a fully
automatic approach like BRT. Additionally, our sequenced modelling procedure
looking for the primary sources of variation, together with the stratified sampling
design, lead us to select a set of lowly correlated predictors for our linear models
(Table S3).

*Geophysical predictors driving SOC*
Considering the difficulties of modelling SOC in a widely heterogeneous mountain
environment (Garcia-Pausas et al., 2017), the Geophysical model provided
important information about SOC drivers in the Pyrenees. TSIS was a key
predictor of SOC with a varying effect depending on macrotopography, slope and
grazing intensity (Table 2). This result contrasts with most of the previous studies
addressing soil carbon in mountain grasslands, which usually pinpoint mean
temperature and precipitation as the most important climate drivers of SOC
(Hobley et al., 2015; Manning et al., 2015; Wiesmeier et al., 2019). Overall, the
TSIS effect on SOC was positive under certain conditions. Sites characterised by
low mean temperatures presented a wider spectrum of TSIS values than warm
sites (Fig. S8). Considering that climate regulates large scale patterns of
aboveground net primary production (Chapin et al., 1987), a positive effect of
TSIS on SOC could be associated with higher biomass accumulation in cold
locations with more favourable temperatures during summer, this fact reducing
geophysical stress for plants (Garcia-Pausas et al., 2007; Kikvidze et al., 2005).



This plant biomass accumulation during summer would overcome an eventual
increase of soil organic matter decomposition rates due to high temperatures
(Sanderman et al., 2003), which could even be diminished if microbial biomass
decreases as a result of soil moisture reduction (Puissant et al., 2018).
The interactions of TSIS with macrotopography and slope illustrate the capacity
of landscape factors to modulate macroclimate effects on soil (Hook and Burke,
2000). Induced microclimate changes are often the explanation for the effects of
topography in SOC (Lozano-García et al., 2016). In our case, SOC stocks
increased with temperature seasonality particularly at mountain exposed areas
(Fig. 3A; Table 2). In protected sites, located in shady slopes and valley bottoms,
the hypothesized positive effect of high TSIS values on productivity could be
mitigated due to lower solar radiation, longer snow-covered periods and freezing
episodes (Garcia-Pausas et al., 2007; López-Moreno et al., 2013). Conversely,
negative effects of low TSIS values on productivity could be compensated thanks
to more humid conditions in protected than in exposed sites (Garcia-Pausas et
al., 2007). Additionally, it is important to take into account that differences in SOC
between exposed and protected sites may also occur through other mechanisms,
for instance the alteration of soil physico-chemical properties like pH, soil texture
or stoniness (Zhang et al., 2018) or differences in vegetation (Sebastià, 2004).
Since we used a hierarchy of controls approach (Manning et al., 2015), these
topography indirect effects could be behind the exclusion on the linear models of
some predictors selected in the BRT model, like silt or pH (Figs. 2 & 3).
In addition, high TSIS values compensated SOC decrease in steep slopes,
probably due to reduced carbon inputs and increased carbon losses induced by
high soil erosion (Yuan et al., 2019 and refferences therein). The decrease in



SOC stocks under low TSIS values were also compensated by grazing pressure
increase (Fig 3D). Recent meta-analyses conclude that grazing has a commonly
decreasing, but strongly context-specific effect on SOC, depending on other
factors like climate, soil type vegetation or grazing intensity (Abdalla et al., 2018;
Eze et al., 2018; Mcsherry and Ritchie, 2013). Particularly, light and medium
grazing intensities can increase SOC inputs by dung deposition and promoting
aboveground and root biomass production (Franzluebbers et al., 2000; Zeng et
al., 2015). Considering that in our semi-natural grasslands all grazing intensities
are relatively low (see methods), our medium and high stock rates may increase
soil carbon inputs in low seasonality locations by enhancing productivity.
Interestingly, clay content and precipitation presented interacting effects on SOC
(Fig. 3C; Table 2). Both MAP and clay content are widely assumed to be
positively correlated to SOC (Wiesmeier et al., 2019). High MAP would increase
SOC inputs by promoting plant productivity (Author et al., 2000; Hobley et al.,
2015). Clay positive effects are often attributed to a larger contact surface of soil
particles (Kennedy et al., 2002), the absorption of negatively charged organic
matter, high soil water retention and the exclusion of decomposer organisms due
to their low pore size (Krull et al., 2001). In our study, high water contents may
inhibit decomposition if a shortage of oxygen supply occurs (Xu et al., 2016b).
However, as MAP values increased, clay effect on SOC became negative. To
explain low SOC values at high MAP and high clay content, McSherry and
Rithchie (2013) hypothesized that finer texture soils could be waterlogged more
frequently, leading to inhibition of root growth and soil C allocation belowground.
*Biochemical predictors driving SOC*





Adding soil nutrient and herbage predictors to our modelling procedure implied
the substitution of the terms including clay content and macrotopography by the
newly added terms (Tables 2 & 3), highlighting the importance of indirect effects
of these variables on SOC through other small scale predictors (Leifeld et al.,
2015; Xu et al., 2016b; Zhu et al., 2019). In this case, we obtained a complex
model with some surprising, less frequently tested effects involving interactions
between graze type, soil nutrients and herbage quality variables (Table 3, Fig 4).
Although our interpretations have limitations because our models were based on
observational data, they can still provide some hints about some of the most
complex and unknown relationships between SOC and its drivers. In can also
contribute to generate testable hypotheses in latter studies.
As expected, SOC increased with the C/N ratio (Fig 4A), which is an indicator of
the difficulty of soil organic matter decomposition by soil microbes, decreasing
decomposition rates of SOC with increasing soil C/N (Wanyama et al., 2019; Xu
et al., 2016b). Conversely, total soil N conditioned livestock type effect on SOC
in a surprising way. Cattle grazed grasslands stored more SOC than mixed and
sheep grazed ones under low soil N conditions, whereas the opposite occurred
at high soil N content (Fig. 4B). Chang et al. (2018) found that in a N poor semi-
arid grassland, sheep decreased SOC content in comparison to cattle due to
vegetation changes caused by their feeding preference for highly palatable forbs,
promoting less palatable grasses which supported less root biomass. A shift
towards higher grass biomass with sheep grazing was also found in the Pyrenees
(Sebastia et al., 2008). Conversely, in our study mixed grazing increased SOC,
probably through effects on soil environment and decomposition processes. Our
results suggested that those processes could vary depending on soil conditions.





Negative effects of sheep grazing on SOC through their selective feeding could
occur mostly in poor N soils (Fig 4B). Under such conditions, palatable plants
could produce higher SOC inputs, since plant productivity is more reliant on the
ability of fixing atmospheric N of legumes (Van Der Heijden et al., 2008) and the
exceptional capacity of forbs to allocate C in roots is especially stimulated (Ågren
and Franklin, 2003; Warembourg et al., 2003). However, these processes could
be different under different soil N conditions, although the concrete mechanisms
are hard to suggest, since livestock type may affect SOC content not only through
changes in plant composition, but other differences in certain features of livestock
assemblages, like trampling, faeces deposition patterns or   effects on plant
regrowth, which could promote differences in soil respiration and/or plant
productivity (Aldezabal et al., 2019; Chang et al., 2018; Liu et al., 2018), resulting
in different SOC levels under different grazers. Grasslands with mixed grazed
regimes stored even more SOC than sheep grazed ones under high soil N
conditions (Fig. 4B, Table 3). This result emphasises that mixed livestock
assemblages deserve particular attention as they can affect plant composition
distinctly from single grazing species regimes or alter traveling and trampling
behaviours of grazing animals (Chang et al., 2018; Liu et al., 2015).
Model terms involving herbage predictors could represent both biochemical and
physical pathways of litter incorporation to soil organic matter (Cottrufo 2015). In
our model, NDF was negatively related to SOC at high N values (Fig 4C). NDF
proportion represents the amount of structural compounds on litter, and hence is
inversely related to non-structural compounds content (Goering and Van Soest,
1970).  The latter are the main source of organic matter formation at the early
stages of decomposition, and they are incorporated into microbial biomass in a





highly efficient way (Cotrufo et al., 2013). However, if microbial necromass is
recycled by microbes before its incorporation to mineral-associated organic
matter, it could be respired and mineralized instead of stored (Córdova et al.,
2018). Thus, our model suggested that incorporation of labile and fast
metabolized non-organic compounds to soil organic matter could be a pathway
of SOC allocation at high soil N in Pyrenean grasslands. At low soil N conditions,
induced changes in microbial composition or priming effects (De Deyn et al.,
2008; Fontaine et al., 2007; Wild et al., 2019; Yan et al., 2018) may disable SOC
accumulation trough this biochemical pathway.

On the other hand, the ADL/NH ratio was positively related to SOC in sheep and
mixed grazed grasslands (Fig. 4D). The ADL/NH ratio is a commonly used
indicator for the resistance of litter to degradation, particularly at later stages of
decomposition (Taylor et al., 1989). Hence, the increase of SOC with ADL/NH
could be related to the physical pathway of soil organic matter incorporation,
forming coarse particulate organic matter (Cotrufo et al., 2015). Moreover, our
model suggests that this pathway would be inhibited under cattle grazing,
presumably because of their less selective diet and higher digestive efficiency
than sheep (Rosenthal et al., 2012; Sebastià et al., 2008). Since lignin content is
inversely related to plant palatability (Moore and Jung, 2001), plants with high
lignin content will be avoided with greater probability under sheep-based
management regimes (Wang et al., 2018), and that would promote differences in
recalcitrant litter mineralization rates. Additionally, lower diet selectivity and
higher digestive efficiency of cattle compared with sheep, can result into less



recalcitrant faeces (Wang et al., 2018), which could explain also SOC differences
between grazer types at high ADL/NH conditions.
*Implications of livestock effects on SOC*
One key point of our results is that they highlight the need for a deeper research
effort in disentangling not only grazing intensity but grazer type effects on
grassland soil organic carbon and nutrient cycling under different environmental
circumstances. Our results concerning interactions between grazer type and
herbage quality provide some evidence of grazing effects not only through
alterations of plant communities that were reported by previous studies in the
region (Canals and Sebastià, 2000; Sebastià et al., 2008),but also through
interactions with them. Although grazing effects were not the most important
factors affecting SOC stocks, this is by far the easiest component to manipulate
in order to increase or maintain SOC in soils and face climate change (Komac et
al., 2014). Despite the need of a precise knowledge on the effects of different
land uses on ecosystems for climate change mitigation (Lo et al., 2015)  studies
addressing grazer type effects on SOC are scarce (i.e. Zhou et al., 2017; Chang
et al., 2018). Considering our results, we would suggest to carry out more
experiments testing the effects of livestock type on SOC under different soil
fertility conditions and plant communities with contrasting herbage quality
parameters.
To conclude, we showed how a combination of regional, landscape,
management, soil properties, soil nutrients and herbage factors might drive SOC
stocks in the Pyrenees. Among all the regional and landscape scale factors, a
seasonality variable, TSIS seemed to be the most decisive, although interacting





with some topographical drivers and grazing intensity. To our knowledge, this is
the first time these factors were combined together with soil nutrients and
herbage quality factors to model SOC. Soil N was a crucial factor modulating the
effect of livestock species and NDF, and herbage recalcitrance effect on SOC
varied depending on grazer species. Our study highlight the need to expand
knowledge about grassland SOC drivers under different conditions, specially
grazing, as this is the most easily tractable factor affecting SOC and it has other
advantages like preventing the accumulation of aboveground C and reducing the
risk of forest fires (Nunes and Lourenço, 2017). We provided the basis to
generate new testable hypothesis for latter studies that may be useful to design
improved policies to palliate climate change.
**DATA ACCESSIBILITY**
Data are not public as the PASTUS database is currently being used for other
research projects. Please contact one of us by e-mail for queries concerning the
data used in this study.
**Acknowledgements**
We would like to express our thanks to the many people who collaborated in
fieldwork, sample processing and data analysis over time. Research in this paper
is based on the PASTUS database, which was compiled from different funding
sources over time, the most relevant being: the EU Interreg III- A Programme
(I3A- 4- 147- E) and the POCTEFA Programme/Interreg IV- A (FLUXPYR, EFA
34/08); the Spanish Science Foundation FECYT- MICINN (CARBOPAS:
REN2002- 04300- C02- 01; CARBOAGROPAS: CGL2006- 13555- C03- 03 and
CAPAS: CGL2010- 22378- C03- 01); the Foundation Catalunya- La Pedrera and



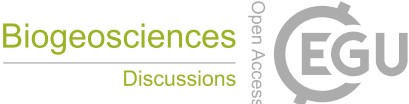

the Spanish Institute of Agronomical Research INIA (CARBOCLUS: SUM2006-
00029- C02- 0). L. San Emeterio was funded through a Talent Recruitment grant
from Obra Social La Caixa - Fundación CAN. ARAID foundation is acknowledged
for support to J.J. Jiménez. This work was funded by the Spanish Science
Foundation FECYT- MINECO (BIOGEI: GL2013- 49142- C2- 1- R) and the
University of Lleida (PhD Fellowship to AR).

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



**Table captions**

Table 1: Considered factors affecting SOC in some recent studies. $V$: the study considers this variable type; $X$: the

study does not consider this variable type.

Table 2: Results of the geophysical model for soil organic carbon ($R^2_{Adj}$ = 0.34).

Table 3: Results of the Combined model for soil organic carbon ($R^2_{Adj}$ = 0.61).

**Figure captions**

Figure 1: Conceptual scheme used in this work to relate potential environmental drivers with

SOC. We assume that drivers may affect soil organic carbon (SOC) both directly or hierarchically

through other driver. Interactions between factors from different types could also drive SOC.

Grazing management has a special location as may act through different paths and interact with

factors at different scales.

Figure 2: Relative contributions (%) of predictor variables in the final BRT model obtained. Soil

N: soil nitrogen; Soil C/N: soil carbon to nitrogen ratio, Clay: clay content; Abiom: aboveground

biomass; ADL: acid-detergent lignin; Loam: loam content; K: soil potassium; TSIS: mean

summer temperature minus mean annual temperature; NDF: neutro detergent fibre; pH: soil

pH; CH: carbon in the herbage; Mg: soil magnesium; Slope: terrain slope; MAP: mean annual

precipitation; ADF: acid detergent fibre. See Table S1 for more details about variables

Figure 3. The relationship between SOC and regional and landscape scale factors in the

Geophysical model. In A) solid lines and circles represent exposed hillsides, and dotted lines and

triangles indicate protected hillsides. In D) solid lines and circles indicate low grazing

management intensity, dotted lines and triangles indicate medium grazing management

intensity and dashed lines and squares indicate high grazing management intensity. In A-D line

and plane values are predictions of the model across the corresponding predictors´ range





according to estimate marginal means. Grey areas around regression lines indicate standard
errors. In A) and D) points indicate actual values.
Figure 4. The relationship between SOC and biochemical and herbage factors in the Combined
model. In B) and D) solid lines and circle points represent cattle-grazing, dashed lines and
square points indicate sheep-grazing and dotted lines and triangle points indicate mixed-
grazing. In A-D line and plane values are predictions of the model across the corresponding
predictors´ range according to estimate marginal means. In A-D line and plane values are
predictions of the model across the corresponding predictors´ range according to estimate
marginal means.  Grey spectrum indicate 95% confidence intervals. In A) and D) points indicate
actual values.





## Tables

Table 1: Considered factors affecting SOC in some recent studies. $V$: the study considers this variable type; $X$: the study does not consider this variable type.

*Fertilizer effects.

** Only aboveground and/or belowground biomass index

| Article | Location | LAT (º) | LONG (º) | MAP (mm) | MAT (°C) | Topography and bedrock | Grazing Management | Soil properties | Soil nutrients | Herbage |
|---|---|---|---|---|---|---|---|---|---|---|
| ***Present study*** | *Pyrenees* | *42.14 – 43.3* | *-1.22 – 2.26* | *964 – 1586* | *1.1 – 9.9* | V | V | V | V | V |
| Duarte-guardia et al., 2019 | Worldwide | -51.72 – 80.23 | -163.95 – 158.25 | 65 – 5115 | -21.2 – 30 | V | X | V | X | V** |
| Abdalla et al., 2018 | Worldwide | -45.85 – 51 | -114 – 120.7 | 150 – 1650 | 0 – 21 | X | V | V | X | V |
| Eze et al., 2018 | Worldwide | -44 – 65 | -121 – 175 | 120 – 2000 | -4.8 – 26.8 | X | V | V | V* | V** |
| Peri et al., 2018 | South Patagonia | - 52 – -45 | -73.5 – 65.5 | 139 – 865 | 4.2 – 11 | V | V | X | X | V |
| Zhang et al., 2018 | Northern China | 103.5 – 124.16 | 32.5 – 42.5 | 500 – 1000 | 8 – 14 | V | V | V | X | X |
| Zhao et al., 2017 | Mongolia | 41.95 – 53.93 | 108.28- 116.2 | 150 – 400 | -1.3 – 2.1 | X | V | V | X | V |
| Zhou et al., 2017 | Worldwide | -42.1 – 52.3 | -121 – 175 | 200 – 600 | 0 – 10 | X | V | X | X | X |
| Deng et al., 2016 | Eastern China | 28.71 – 30.45 | 120.87 – 122.43 | 940 – 1720 | 16.86 – 18.57 | V | X | V | X | X |
| Gray et al., 2015 | Eastern Australia | -16.7 – -43.5 | -31.8 – -28.7 | 500 – 2000 | 10 – 30 | V | X | X | X | V |
| Lu et al., 2017 | Qinghai-Tibetan Plateau | 27 – 32 | 83 – 108 | 37 – 718 | -4.04 – 6.3 | X | V | X | X | X |
| Chang et al., 2015 | Tibet | Not Reported | Not Reported | 397 – 1910 | 1.7 – 15.5 | V | X | X | X | V |
| Manning et al. 2015 | England | 50.77– 54.58 | -4.43 – 0.87 | 596 – 3191 | 6.5 – 10.9 | X | V | V | X | V |
| McSherry & Ritzie 2013 | Worldwide | Not reported | Not reported | 180 – 950 | Not reported | X | V | V | X | V |
| Garcia-Pausas et al. 2007 | Pyrenees | -7 – 2.2 | 42.5 – 42.9 | 1416 – 1904 | -0.7 – 5 | V | X | V | X | X |






Table 2: Results of the geophysical model for soil organic carbon ($R^2_{Adj}$ = 0.34).

| Model term | Estimate | SE | t-value | P-value | |
|---|---|---|---|---|---|
| Intercept | -0.525 | 1.802 | -0.291 | 0.771 | |
| **Climate variables** | | | | | |
| MAP | 0.003 | 0.001 | 4.560 | <0.001 | *** |
| TSIS | -0.098 | 0.228 | -0.429 | 0.669 | |
| **Topography variables** | | | | | |
| Slope | -0.339 | 0.095 | -3.569 | 0.001 | *** |
| Exposed | -3.130 | 0.936 | -3.344 | 0.001 | ** |
| **Soil type variables** | | | | | |
| Clay | 0.121 | 0.027 | 4.500 | <0.001 | *** |
| **Management variables** | | | | | |
| Low intensity | -5.013 | 1.196 | -4.192 | <0.001 | *** |
| Medium intensity | 2.012 | 1.168 | 1.722 | 0.088 | |
| **Interactions between variable types** | | | | | |
| TSIS x Exposed | 0.417 | 0.124 | 3.358 | 0.001 | ** |
| TSIS x Slope | 0.044 | 0.013 | 3.452 | 0.001 | *** |
| MAP x Clay | 0.000 | 0.000 | -4.637 | <0.001 | *** |
| TSIS x Low intensity | 0.655 | 0.159 | 4.110 | <0.001 | *** |
| TSIS x Medium intensity | -0.262 | 0.156 | -1.684 | 0.095 | |







Table 3: Results of the Combined model for soil organic carbon ($R^2_{Adj}$ = 0.61).

| Model term | Estimate | SE | t-value | P-value | |
|---|---|---|---|---|---|
| Intercept | -0.290 | 1.458 | -0.199 | 0.843 | |
| **Climate variables** | | | | | |
| MAP | -0.001 | 0.000 | -2.434 | 0.017 | * |
| TSIS | -0.004 | 0.181 | -0.022 | 0.982 | |
| **Topography variables** | | | | | |
| Slope | -0.225 | 0.078 | -2.868 | 0.005 | ** |
| **Management variables** | | | | | |
| Cattle | 0.487 | 0.101 | 4.834 | <0.001 | *** |
| Mixed | -0.289 | 0.093 | -3.106 | 0.002 | ** |
| Low intensity | -3.249 | 1.014 | -3.204 | 0.002 | ** |
| Medium intensity | 1.666 | 1.073 | 1.553 | 0.123 | |
| **Soil nutrient variables** | | | | | |
| Log(Soil C/N) | 0.665 | 0.076 | 8.777 | <0.001 | *** |
| Soil N | 3.302 | 0.617 | 5.349 | <0.001 | *** |
| **Herbage variables** | | | | | |
| NDF | 0.014 | 0.006 | 2.525 | 0.013 | * |
| Herbage ADL/NH | 0.026 | 0.009 | 2.987 | 0.003 | ** |
| **Interactions between variable types** | | | | | |
| TSIS x Slope | 0.030 | 0.010 | 2.833 | 0.006 | ** |
| TSIS x Low intensity | 0.423 | 0.136 | 3.104 | 0.002 | ** |
| TSIS x Medium intensity | -0.214 | 0.143 | -1.495 | 0.138 | |
| Soil N x Cattle grazing | -0.736 | 0.168 | -4.380 | <0.001 | *** |
| Soil N x Mixed grazing | 0.493 | 0.175 | 2.813 | 0.006 | ** |
| Soil N x NDF | -0.039 | 0.011 | -3.505 | 0.001 | *** |
| Cattle x Herbage ADL/NH | -0.030 | 0.010 | -2.872 | 0.005 | ** |
| Mixed x Herbage ADL/NH | 0.014 | 0.011 | 1.252 | 0.213 | |








**Figures**

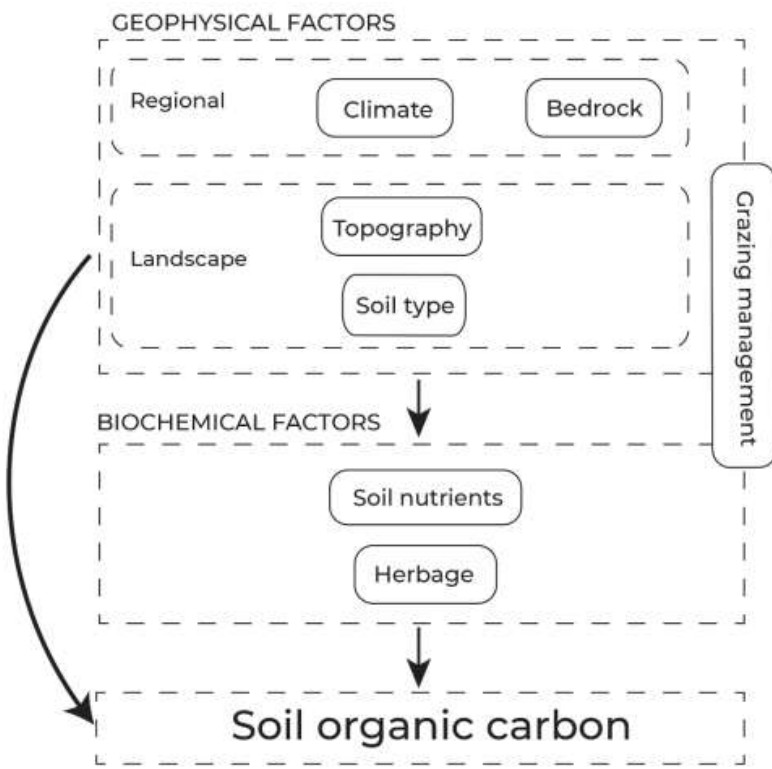


Figure 1: Conceptual scheme used in this work to relate potential environmental drivers with
SOC. We assume that drivers may affect soil organic carbon (SOC) both directly or hierarchically
through other driver. Interactions between factors from different types could also drive SOC.
Grazing management has a special location as may act through different paths and interact with
factors at different scales.




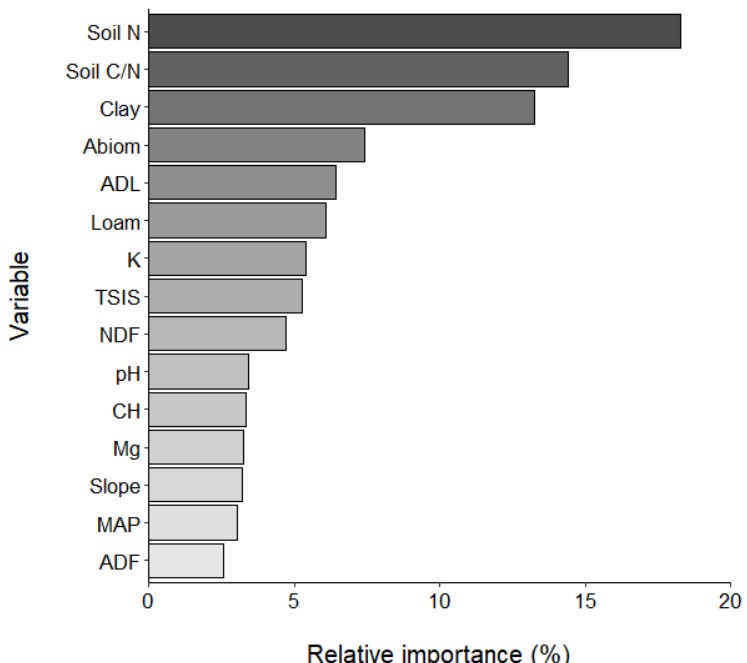

Figure 2: Relative contributions (%) of predictor variables in the final BRT model obtained. Soil
N: soil nitrogen; Soil C/N: soil carbon to nitrogen ratio, Clay: clay content; Abiom: aboveground
biomass; ADL: acid-detergent lignin; Loam: loam content; K: soil potassium; TSIS: mean
summer temperature minus mean annual temperature; NDF: neutro detergent fibre; pH: soil
pH; CH: carbon in the herbage; Mg: soil magnesium; Slope: terrain slope; MAP: mean annual
precipitation; ADF: acid detergent fibre. See Table S1 for more details about variables.



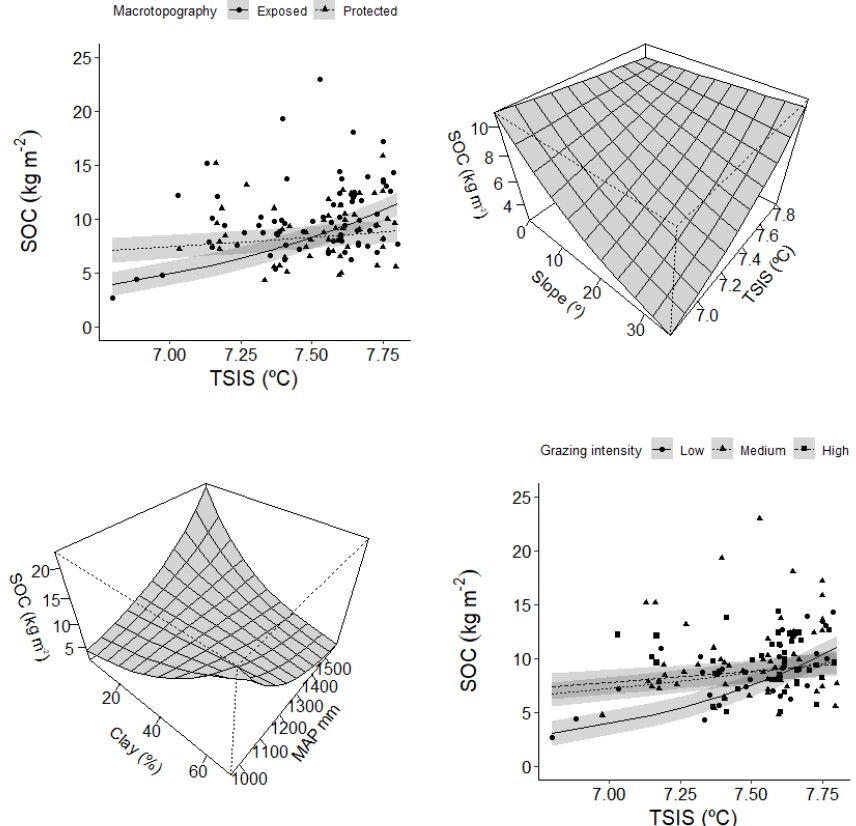

Figure 3. The relationship between SOC and regional and landscape scale factors in the Geophysical model. In A) solid lines and circles represent exposed hillsides, and dotted lines and triangles indicate protected hillsides. In D) solid lines and circles indicate low grazing management intensity, dotted lines and triangles indicate medium grazing management intensity and dashed lines and squares indicate high grazing management intensity. In A-D line and plane values are predictions of the model across the corresponding predictors´ range according to estimate marginal means. Grey areas around regression lines indicate standard errors. In A) and D) points indicate actual values.





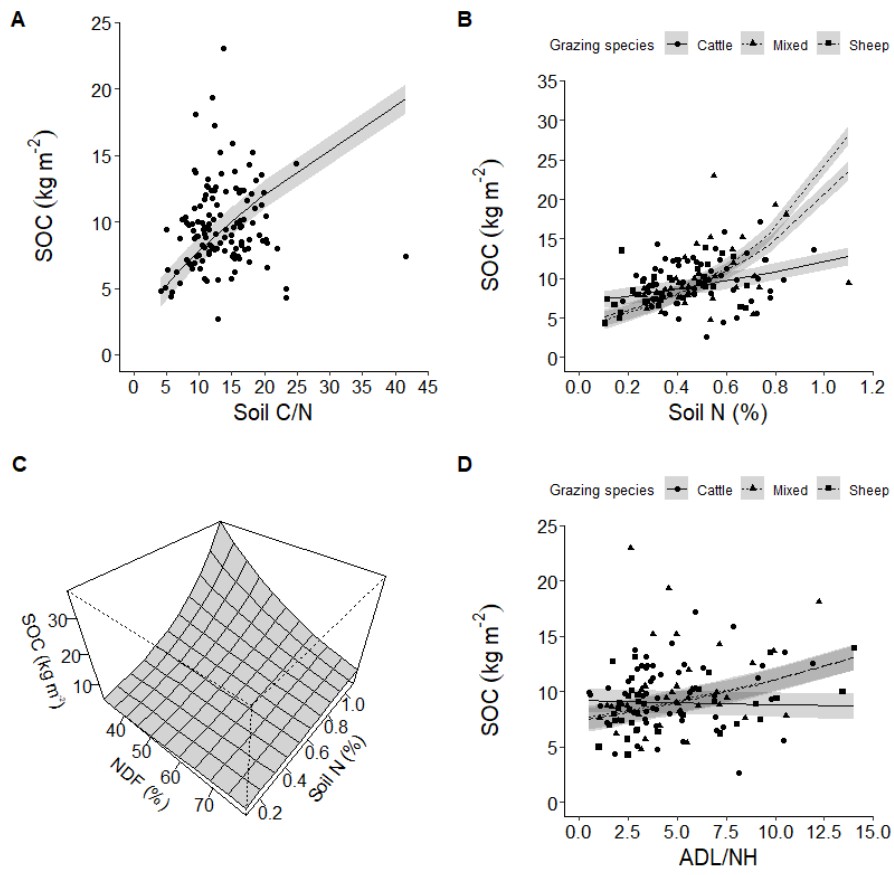


Figure 4. The relationship between SOC and biochemical and herbage factors in the Combined
model. In B) and D) solid lines and circle points represent cattle-grazing, dashed lines and
square points indicate sheep-grazing and dotted lines and triangle points indicate mixed-
grazing. In A-D line and plane values are predictions of the model across the corresponding
predictors´ range according to estimate marginal means. In A-D line and plane values are
predictions of the model across the corresponding predictors´ range according to estimate
marginal means.  Grey spectrum indicate 95% confidence intervals. In A) and D) points indicate
actual values.