# Peer review of "Interactions between biogeochemical and management factors explain soil"

_Biogeosciences, 2020_

## Referee Comment (RC1) · Anonymous Referee #1 · 27 May 2020

Overall, the manuscript entitled 'Interactions between biogeochemical and management factors explain soil organic carbon in Pyrenean grasslands' would have potential to be of great interest for the readers of Biogeosciences Journal. It provides interesting results on the effect of different drivers on soil carbon stocks in Pyrenean grasslands. However, I have noticed some important points that need to be addressed before this manuscript can be considered for publication.

Concerning the abstract, I think that the scope and objectives of the study need to be better defined. After reading it, we do not have a clear idea of what factors have been tested. I have the same feeling after reading the introduction. Overall, we understand

that there are many factors which can influence soil C stocks at different scales, but it is difficult to understand what are the real objectives of the study. Is the objective to determine which factors influence the most the soil C stocks, is this analysis done for different scales? In the material and methods section, the main issue that I noticed concerns the statistical approach. It is not clear for me why two separate approaches were done. It adds a certain complexity to the article and it needs to be better presented according to the objectives for each approach. Are both the approaches really relevant for the paper? The links between the objectives and the chosen modelling approach needs to be better defined. Also, concerning the calculation of soil C stocks, it would have been appropriate to correct soil C stocks according to the equivalent soil mass approach to account for possible differences in bulk density values (Ellert and Bettany, 1995; Ellert et al., 2008). Concerning results and discussion, even if the ideas are, overall, well supported by relevant references and the limits are underlined, I think that the organization will be improved after the clarification of the objectives and the corresponding analyses. Also I noticed repetitions of results in the 'results' section and in the 'discussion' section so I would suggest to group all the results and discussion in one section if the journal guidelines allow it. Finally, it would be important that the manuscript be reviewed for the English. Some corrections might be necessary.

In the next paragraph, I developed some detailed comments that will help the authors to improve the manuscript.

L 53-54 "at small spatial scales" instead of "at detailed spatial scales" L 56-57 I am not sure that it is a good reason to do a study... What is the objective of the study by using this set of data? L 58 Do the authors have an explicative purpose or a predictive purpose ? That is not clear for me, as they also use the 'predictors' term. L 59 This factor should be better defined. L 65 I think that the coma is not necessary. L. 95-96 I think that these variables should be better described. Also "be" should be removed. These factors are not studied or they are not factors with a relevant impact in other studies? L.112 Same question than earlier: are they omitted because they do not

impact the SOC stocks? L. 113 "focusing" instead of "focus" L. 116 Overall, for the whole manuscript, the authors need to specify if it is SOC stock or concentration. L. 127 What type of management do you consider? L. 136 And what was their conclusion in regards of your objectives? L. 140 Among which drivers? There are many factors that can interact or be correlated together. We need to know which drivers will be tested. The authors should be clearer on the objectives of this study. L. 141 Âń assess Âż L. 151-153 Do the authors want to study the effects of various factors, their links between them, the importance of the factors...? L. 175 Âń grazer type Âż instead of Âń grazing management Âż, no ? L 189-190 Are the soil samples from the 4 quadrats composited to form one soil sample per depth for each grassland patch? L. 192-193 I think this paragraph should appear before... L. 194 There should be a coma between landscape and livestock L. 199 But you don't speak of mean summer temperature before. . . L. 200-201 How did you appreciate that? We need to have more details on this factor. L. 218 For each patch considered? L. 229 For determination of bulk density? L. 233-234 This sentence is not clear. L. 243 It should have been important to correct soil C stocks according to the equivalent soil mass approach. L. 249 What was the vegetation : grassland species etc. L. 267 The size of the police is not the same for all this paragraph. Does this paragraph of NIRS analysis refer to the analyses presented in the previous paragraph? It is not clear. L. 293 Among which variables? L.301 "Firstly" instead of "First" L. 306-307 What is this new set of variables? L. 314-316 Why choosing these two models, on which hypothesis did you decide these two groups? L. 316-320 Maybe it should be more appropriate in the introduction... L. 374 Why there are not all the predictors described in the introduction in this model? Grazing management for example? L.381 Why these two variables are not selected in the model? L.411 Some repetition from the results section... L.444-447 I wonder if the BRT model is really relevant for the manuscript. . . Also, Are you sure it is table S3 ??? L. 487 SOC decrease with increase of slope L.489 Not clear. . . L.491 What I see is that SOC stocks are lower under low intensity of grazing for low values of TSIS. . . L.494-499 It is not really clear. L.507 high soil water contents? L.525 "which might be

explained by" instead of "which is an indicator"

Ellert, B.H., Bettany, J.R., 1995. Calculation of organic matter and nutrients stored in soils under contrasting management regimes. Can. J. Soil Sci. 75, 529–538. Ellert, B.H., Janzen, H.H., VandenBygaart, A.J., Bremer, E., 2008. Measuring change in soil organic carbon storage. In: Carter, M.R., Gregorich, E.G. (Eds.), Soil Sampling and Methods of Analysis. CRC Press Taylor & Francis Group, Boca Raton, FL, pp. 25–38.

---

## Referee Comment (RC2) · Anonymous Referee #2 · 15 Jun 2020

General comments The manuscript aims to understand how environmental and management factors affect SOC in mountain grasslands. And fitted a set of models with explicative purposes using data that comprise a wide range of environmental and management conditions to find the most important driver of grassland SOC. The authors are to be commended on the framing of an interesting study, the collection of a reasonable set of ancillary environment and management data and soil data in what appears to be good quality piece of research. The workload of this article is very huge. However, too many sections and repetitive statements in this article. Be better structured and more concise to attract readers. Deep discussion and comparison of your work is needed in an international context. In discussion section, some discussion on the

mechanism of environmental and management factors should be added. I suggest you add a conclusion section, a concise and clear conclusion will make your article more eye-catching and let readers understand the conclusion of this article more quickly and easily. As the manuscript contains some uncertainties in description of the methods, results, and English writing, I suggest a moderate revision necessary before it can be acceptable for publication in this journal.

Specific comments Line 75 "Soil organic carbon plays key roles in the terrestrial ecosystems." It sounds strange.

Line 179 At least one to two replicates of each patch type were sampled. What are the types of the patch?

Line 155 Not clear sampling design description. Showing a figure with sampling design would be helpful. Add a schematic of experimental design to make it clearer.

Line 192 The abbreviation for soil organic carbon had appeared in line 75, here only need to write SOC.

Line 193 There are 30 variables written in table S1, but here you have written 29 independent environmental variables. Are the two management variables belong to environmental variables? Please check these numbers.

Line 194 These variables were grouped into Regional, landscape, livestock management, soil nutrient stocks, and herbage variables? If so, replace ":" with ",".

Line 201 MTS?

Line 220 Here used livestock stocking rates which measured as livestock units ha-1 to determine grazing intensity. But the feed intake of different types of livestock is different. For example, the intake of cattle is higher than that of sheep. So, can't simply use the livestock units/ha-1 as livestock stocking rates, you need use standard livestock unit.

Line 314 Geophysical model based on geophysical predictors and grazing management? There haven't grazing management in Figure S4.

Line 371 Authors need to better describe statistics of SOC.

Line 375 Generally, a part of the sample is used for modeling, and the other part is used for validation. Please describe clearly in here and in Line 279.

Line 379 Silt in here, loam in fig.2. Use consistent terminology of silt, loam, etc? Use one, Please!

Line 382 Why TSIS was the most relevant selected climate predictor? In figure 6s, Soil C/N has a higher relative importance.

Line 383 Please confirm this sentence and the quoted figure. I didn't find TSIS in figure S5 and S6.

In table s1, TSIS described as MST-MAT. In figure s8, MMT also described as MST-MAT Use consistent terminology of MMT, TSIS, etc? Use one, Please!

Line 381 Aboveground biomass and silt had a high relative contribution in the final BRT model obtained, why not selected them in the linear models?

Line 1121 Please add the fitting equation in figure 3 and 4. It is hard to distinguish which trend line belongs to which grazing species or grazing intensity. You can distinguish by color, or add the legend.

Line 25 in SUPPLEMENT Figure S1: points indicate sampling location, sampling location means the sample patches? Please add the legend of the points in this figure.

Line 39 in SUPPLEMENT There is no reference of Figure S3 in the text.

---

## Author Comment (AC1) · 17 Jul 2020

Dear referee 1,

Please find in your following lines and attached in a .pdf file the answers to your revisions. We think that .pdf file will be easier to read.

Referee 1 Overall, the manuscript entitled 'Interactions between biogeochemical and management factors explain soil organic carbon in Pyrenean grasslands' would have potential to be of great interest for the readers of Biogeosciences Journal. It provides interesting results on the effect of different drivers on soil carbon stocks in Pyrenean

grasslands. However, I have noticed some important points that need to be addressed before this manuscript can be considered for publication. We really appreciate your revision. Your comments definitely will improve our manuscript. Concerning the abstract, I think that the scope and objectives of the study need to be better deffned. After reading it, we do not have a clear idea of what factors have been tested. I have the same feeling after reading the introduction. Overall, we understand that there are many factors which can influence soil C stocks at different scales, but it is difffcult to understand what are the real objectives of the study. Is the objective to determine which factors influence the most the soil C stocks, is this analysis done for different scales? We have revised the abstract and the introduction sections, following your specific comments. Under our view, the scope and objectives are now more understandable. In a nutshell, the scope is to study the relative effects, including interaction effects, of geophysical and biochemical SOC drivers, and also to pinpoint how grazing management regulates the effects of other SOC controls. In the material and methods section, the main issue that I noticed concerns the statistical approach. It is not clear for me why two separate approaches were done. It adds a certain complexity to the article and it needs to be better presented according to the objectives for each approach. Are both the approaches really relevant for the paper? The links between the objectives and the chosen modelling approach needs to be better deffned. Also, concerning the calculation of soil C stocks, it would have been appropriate to correct soil C stocks according to the equivalent soil mass approach to account for possible differences in bulk density values (Ellert and Bettany, 1995; Ellert et al., 2008). We explained in the specific comments why we think both statistical approaches are complementary and important. We revised the manuscript to emphasize and make clear this point. However, if our arguments neither convince nor the editors nor the referees, we are open to put the BTR model in supplementary material or even suppress it completely. Note that although it has been argued that the usefulness of using both approaches was not clear, referee 1 made several specific questions about the differences between their results, precisely about the points we consider interesting. We

also commented the point about fixed mass approach for calculating SOC stocks in the corresponding specific comment. Concerning results and discussion, even if the ideas are, overall, well supported by relevant references and the limits are underlined, I think that the organization will be improved after the clarification of the objectives and the corresponding analyses. Also I noticed repetitions of results in the 'results' section and in the 'discussion' section so I would suggest to group all the results and discussion in one section if the journal guidelines allow it. We think separate sections for results and discussion are important, since this is useful for separating the raw statistical results from results discussion and interpretations. We truly believe that the manuscript is going to be easier to read and understand if we maintain this structure. The statistical methods presented here could seem complex, and reading the results separately could help to their understanding since is the shortest and simplest section. Anyway, we followed your advice and we revised the manuscript to make it less repetitive. Your specific comments where greatly valuable for this task. The most important change is we suppressed the first paragraph in the discussion section, which was actually a summary of the result section. We also revised the paragraph about the modeling procedure, and we believe now is more clear. We think the rest of the subsection titles in the discussion were useful to structure the text. Under our view, every sub-section was justified. However, we grouped the second and third subsections (Geophysical, biochemical and grazing management factors driving SOC stocks) as both referees asked us to reduce the number of sections. The idea is that first section gives an idea of the right way of interpreting the models. The second section answers the questions formulated at the end of the introduction; 1: "what are the relative and interaction effects of the geophysical and biochemical SOC controls?" and 2: "How grazing management regulate the effects of other SOC drivers? The fourth subsection discuss particular implications of our results in grazing management, a point which we consider noticeably important. Finally, we separated and revised the conclusion section following the indications of referee 2. Of course, if after this revision, the referees and the editor consider that results and discussion section must be combined, we could do it without a

problem. Finally, it would be important that the manuscript be reviewed for the English. Some corrections might be necessary. We did so. In the next paragraph, I developed some detailed comments that will help the authors to improve the manuscript.

L 53-54 "at small spatial scales" instead of "at detailed spatial scales". Change done. (L54) L 56-57 I am not sure that it is a good reason to do a study... What is the objective of the study by using this set of data? To clarify this, we rewrote this sentence as follows: "Taking advantage of the high variety of environmental heterogeneity in the Pyrenees, we built a dataset that comprise a wide range of environmental and management conditions." (L56) L 58 Do the authors have an explicative purpose or a predictive purpose? That is not clear for me, as they also use the 'predictors' term. The study has an explicative purpose. We have changed "predictors" by "drivers" or "factors" in all the text to avoid misinterpretations. L 59 This factor should be better defined. We specified it in the following way: "We found that temperature seasonality (difference between mean summer temperature and mean annual temperature; TSIS) was the most important geophysical driver of SOC in our study." (L 60) L 65 I think that the coma is not necessary. The comma was removed. (L 67) L. 95-96 I think that these variables should be better described. Also "be" should be removed. These factors are not studied or they are not factors with a relevant impact in other studies? This phrase was rewritten as follows, to clarify that these factors were not even considered in these previous studies and the meaning of climate seasonality: "However, climate regular annual variations, represented by seasonality variables, are be commonly neglected when considering possible SOC drivers affecting SOC in broad-scale models, in spite of being some important factors for plant primary production or enzymatic activity of soil microorganisms." (L 97 - 99) L.112 Same question than earlier: are they omitted because they do not impact the SOC stocks? The phrase was rewritten as follows: However, these variables are commonly omitted as possible drivers of SOC in the broad-scale SOC studies, especially if those focusing on predictive models instead of explicative ones" (L 115 - 116) L. 113 "focusing" instead of "focus" Change done. (L 116) L. 116 Overall, for the whole manuscript, the authors need to specify if it

is SOC stock or concentration. In is SOC stocks. That is now specified in multiple parts of the text. L.127 What type of management do you consider? In our case (natural grasslands), we consider livestock management. However, according to the cited article (Wiesmeier et al., 2019), management effects on grassland SOC in general are poorly understood. We rephrase the sentence: "Apart from these factors, livestock management effects on grassland SOC is..." to clarify that in this paper we only refer to livestock management, which is from far the main management done in natural grasslands." (L 130). L. 136 And what was their conclusion in regards of your objectives? The conclusion in regards of our objectives was that grazing must be considered as a variable that can interact with many variables at multiple scales (as it is represented in Fig. 1). We reordered this paragraph and completed this particular sentence to clarify this point: "It is known that herbivores can affect SOC through different paths, such as regulating the quantity and quality of organic matter returned to soil (Bardgett and Wardle, 2003), or affecting soil respiration and nutrients by animal trampling or soil microbiota alteration (Lu et al., 2017). Moreover, several studies describing interactions of grazing with other SOC drivers at diverse scales have been published (Abdalla et al., 2018; Eze et al., 2018; Lu et al., 2015, 2017; Zhou et al., 2017). Hence, grazing management on grasslands may be considered a unique SOC driver, because it has effects at multiple levels of the driver hierarchy (Fig. 1), both affecting other SOC drivers and interacting with them. However, most of the studies investigating grazing effects on SOC focus on grazing intensity, in spite of evidence pointing to a greater role of grazer species in determining vegetation and SOC (Chang et al., 2018; Sebastia et al., 2008)." (L 131-143) L. 140 Among which drivers? There are many factors that can interact or be correlated together. We need to know which drivers will be tested. The authors should be clearer on the objectives of this study. We modified this paragraph as follows: "In this study, our goal was to identify the main drivers of SOC stocks and their interactions on Pyrenean mountain grasslands. For this purpose, we considered a wide set of regional, landscape, soil geophysical and biochemical and herbage quality factors, together

with grazing management factors. Mountain grasslands comprise a wide range of all these conditions that make carbon stocks highly variable (Garcia-Pausas et al., 2007, 2017). For this reason data analyzed here comprise a wide range of environmental conditions, comparable to studies on SOC developed at continental or even worldwide scales (Table 1). Additionally, we consider an exceptionally broad compilation of drivers (Table 1). To deal with correlations and interactions between SOC drivers, we developed an exhaustive modelling approach based on the controls over function hypothesis (de Vries et al., 2012)." (L. 145 - 155) L. 141 To asses This sentence was changed and this word does not appear. 151-153 Do the authors want to study the effects of various factors, their links between them, the importance of the factors...? We rewritten the questions as follows to put that point clear: "1) what are the relative and interaction effects of the geophysical and biochemical SOC controls? 2) How grazing management regulate the effects of other SOC drivers?" (L 162) L.175 grazer type instead of grazing management Change done L 189-190 Are the soil samples from the 4 quadrats composited to form one soil sample per depth for each grassland patch? Yes, they are. Following the advice of referee 2, we made many clarifications about sampling design, including a new supplementary figure (Fig. S2). L. 192-193 I think this paragraph should appear before... We appreciate this comment, and we also recon that this paragraph could appear at the beginning of the methods section. However, we still find clearer to explain first how the sampling was performed and second how the samples were processed in order to get the environmental variables. L. 194 There should be a coma between landscape and livestock Change done (L 208) L. 199 But you don't speak of mean summer temperature before... We added MST where climate variables were introduced, in the second paragraph of the section 2.2: Regional and Landscape environmental drivers: "Regional variables included climate variables and bedrock. Climate variables were determined from Worldclim 2.0 (Fick and Hijmans, 2017). We selected Mean Annual Temperature (MAT), Mean Summer Temperature (MST), Mean Annual Precipitation (MAP) and Mean Summer Precipitation (MSP)." (L 210 - 213) L. 200-201 How did you appreciate that? We need

to have more details on this factor. During preliminary modelling exercises, two climatic variables appeared repeatedly in all tested models, always significantly contributing to the models with coefficients of similar magnitude but opposite signs: mean annual precipitation and mean summer temperature. In those initial models, even when those two variables were included in interaction with other drivers, this pattern was maintained; that is, the interactions were of opposite sign and the coefficients of the interactions were similar in magnitude. In this way, this is how the TSIS index initially emerged. This index has been found a significant driver of different variables of interest in the PASTUS database, including SOC and plat diversity (Rodríguez et al. 2018). L. 218 For each patch considered? Those grasslands are usually managed communally, and the livestock type and units are based on the number of animals, and type, sent to graze a given area during the grazing season. The unit area is usually related to the municipalities, although this situation might change a little depending on the mountain range. Grazing in the high-altitude grasslands in the Pyrenees is usually free-range. L. 229 For determination of bulk density? Yes. We modified the sentence as follows to clarify this point: "Soil samples were air-dried and weighted to obtain its bulk density. " (L 244) L. 233-234 This sentence is not clear. We rephrased this sentence to clarify it: "We combined 0-10 and 10-20 cm values for obtaining the whole top 20 cm soil layer." (L 249 - 250) L. 243 It should have been important to correct soil C stocks according to the equivalent soil mass approach. We decided to use a fixed depth approach for calculating SOC stocks due to the following reasons. First, the main advantage of fixed mass approaches is that they account and correct differences in bulk density due to temporal changes or when comparing different land uses (Haden et al. 2020). We do not consider variations in time, and neither have contrasting management regimes, as mentioned in the title of Ellert & Bettany's paper (1995). We highlight that in our work samples came from natural mountain grasslands, where grazing intensity is always low to moderate, and moreover, herbivore presence is seasonal. Therefore, we do not expect important changes in bulk density due to land use. Second, we always used the same methods in our samplings (so we could not
take advantage of fixed mass approaches for correcting biases due to different probe diameters, as suggested by Sharma et al. (2020). Finally, fixed mass approaches often have more technical difficulties than fixed depth measures even in the most modern procedures (Haden et al. 2020). On the other hand, Rovira et al. (2015) proposed a fixed mass approach which, as expected, was found to deal properly with bulk density changes but not with stoniness differences. We did not find any other reference dealing with this point. To clarify this point, we added the following lines to the text: "Soil organic carbon (SOC) stocks in the upper 20 cm soil layer were then estimated taking into account the organic C concentration in the sample and its bulk density, and subtracting the coarse particle (> 2 mm) content, following García-Pausas et al. (2007). Despite recent studies suggested that fixed mass SOC stocks estimates are preferable to fixed depth methods because they would be more robust to temporal and land use changes in bulk density (Ellert & Bettany 1995), we chose a fixed depth method for measuring SOC stocks. This decision was based on the fact that our work samples came from natural mountain grasslands, where grazing intensity is always low to moderate, and moreover, herbivore presence is seasonal. Therefore, we do not expect important changes in bulk density due to land use. Additionally fixed mass approaches presented the disadvantages of implying more technical difficulties than fixed depth measures, even in the most modern procedures (Haden et al. 2020), and could not deal well with differences in stoniness." (L 264 - 273) L. 249 What was the vegetation: grassland species etc. We added the following paragraph to provide that information: "Almost all of the plant species in the grasslands from the PASTUS database are perennial (Sebastià, 2004), and plant biodiversity is highly heterogeneous as are the environmental conditions (Rodríguez et al., 2018)." As this is not bromatological information we added this paragraph in lines 178 - 179 where describing the sample site conditions. L. 267 The size of the police is not the same for all this paragraph. Does this paragraph of NIRS analysis refer to the analyses presented in the previous paragraph? It is not clear. We changed some sentences in these paragraphs so now the relationship between these two methods

is clear. Basically, bromatological analysis were done for training NIRS models and getting the remaining values using NIRS spectrum. (L 275) L. 293 Among which variables? Among SOC and all the considered drivers. To clarify, we modified the sentence as follows: "Including all potential SOC drivers, we fitted a model with BRT to identify the most important ones affecting SOC." (L 321) L.301 "Firstly" instead of "First" Change done. (L 330) L. 306-307 What is this new set of variables? This is standard procedure, according to (Elith et al. 2008). As it is explained, it refers to the variables that improve BTR model performance, the model set showed in Fig. 2. L. 314-316 Why choosing these two models, on which hypothesis did you decide these two groups? The geophysical variables are those commonly used in the literature, and are the first source of variation according to the hierarchy of controls over function hypothesis (Manning et al., 2015). Choosing these two models allows us to discuss the effects of geophysical variables on SOC without deleting some effects because of the inclusion of other variables (especially soil nutrients) whose effects may include those of geophysical variables, because geophysical variables could act trough other variables at smaller spatial scales (in this case, the biochemical variables). We consider Geophysical Model is interesting for discussion, since it allow comparisons with previous literature. Additionally, we reported which terms of the Geophysical Model were substituted by the biochemical variables, which suggests that those effects could affect SOC trough biochemical variables, while the other effects probably acted trough other mechanisms too. Finally, we believe that Geophysical Model has interest for future studies aiming to predict SOC in similar environmental conditions. As we mentioned before, these studies usually use what we call here geophysical variables, because they are easy and cheap to measure or obtain (Manning et al., 2015). We modified the referred sentence as follows, to emphasize some of these points: "We built two models (Fig. S4), one model only based on geophysical drivers and grazing management (Geophysical Model), and another model by adding to the former the biochemical drivers: soil nutrients and herbage quality drivers (Combined Model). With this approach we aim to avoid ignoring significant effects of the geophysical variables,

the original source of variation of SOC stocks according to the hierarchy of controls over function hypothesis (Manning et al., 2015), by masking them with the inclusion of biochemical drivers." (L 343 - 349). We also added the following modifications in the "Geophysical drivers driving SOC stocks" subsection of the discussion section to a better explanation of the usefulness of the Geophysical model: "Considering the difficulties of modelling SOC in a widely heterogeneous mountain environment (Garcia-Pausas et al., 2017), the Geophysical Model provided important information about SOC drivers in the Pyrenees. This information could be useful not only for a better understanding of SOC patterns in mountain grasslands, but to future modelling studies aiming to predict SOC, since geophysical variables are easier and cheaper to obtain and measure compared do biochemical ones (Manning et al., 2015)." (L 470 – 476) L. 316-320 Maybe it should be more appropriate in the introduction... We think this is appropriate for the methods as it contributes to the understanding of the modeling procedure. However, we modified the last paragraph of the introduction, to specify these aspects too. In overall we believe that this important paragraph has been widely improved thanks to your comments and suggestions. This text is now as follows: "In this study, our goal was to identify the main drivers of SOC stocks and their interactions on Pyrenean mountain grasslands. For this purpose, we considered a wide set of regional, landscape, soil geophysical and biochemical and herbage quality factors, together with grazing management factors. Mountain grasslands comprise a wide range of all these conditions that make carbon stocks highly variable (Garcia-Pausas et al., 2007, 2017). For this reason data analysed here comprise a wide range of environmental conditions, comparable to studies on SOC developed at continental or even worldwide scales (Table 1). Additionally, we consider an exceptionally broad compilation of drivers (Table 1). To deal with correlations and interactions between SOC drivers, we developed an exhaustive modelling approach based on the controls over function hypothesis (de Vries et al., 2012). To facilitate the formulation of our hypothesis, we classified SOC drivers into three main groups (Fig. 1): i) geophysical factors, which include regional and landscape factors and are

supposed to be the first sources of variation, ii) biochemical factors, which include soil nutrients and herbage factors and could be conditioned by geophysical factors, and iii) grazing management factors, which could affect SOC through multiple interactions with the rest of the variables at multiple scales. In particular, the specific questions of this study are 1) what are the relative and interaction effects of the geophysical and biochemical SOC controls? 2) How grazing management regulate the effects of other SOC drivers?" (L144 - 164) L.374 Why there are not all the predictors described in the introduction in this model? Grazing management for example? Because they were discarded in the BTR modelling procedure. Note that three-based methods can have difficulties in modeling some functions (Elith et al. 2008). We answered about why using both methods three comments below. L.381 Why these two variables are not selected in the model? This point was discussed in the "Considerations about the modelling procedure" subsection inside the discussion section. "As a regression tree machine learning technique, the BTR model identified a set of SOC stocks predictor drivers (Fig. 2) avoiding some of the linear model disadvantages, like guarding against the elimination of good predictor drivers correlated to others or automatically modelling non-linear effects (Cutler et al., 2007; Elith et al., 2008).Thus, the BRT model included some SOC predictors, like a positive logarithmic like effect of aboveground biomass or soil K on SOC (Fig. S7), which could be masked by the effects of other variables in our linear models (Yang et al., 2009)." Basically, multiple predictor variables can not only be correlated but also have true cause-effect relationships between them (i.e. precipitation and aboveground biomass), what means that in a linear model, some drivers could be discarded not because they have no effects on the response variable, but because their effects were already included in other variable. In other words, some variables, like aboveground biomass, soil K or silt were not included in the linear models probably because they were correlated with other drivers which were included in the models. The advantage of including BTR analysis is that we could detect some of these variables. L.411 Some repetition from the results section... We deleted this paragraph as it is repetitive. L.444-447 I wonder if the BRT model

is really relevant for the manuscript. . . Also, Are you sure it is table S3??? It is table S5 (change done). To summarize, the BTR model is relevant insofar it provides information about the effects of the variables not included in the linear models due to correlation. Is this information relevant enough for the manuscript? We think it is. For instance, if BTR model were not included, one question one referee or regular reader would ask would be: "How you can explain that aboveground biomass was not included in your models?" "Does it means that aboveground biomass had no effects on SOC?" The answer is that aboveground biomass had effects on SOC, but in the GLMs these effects were masked by other variables which explain more variation than aboveground biomass, and probably affect SOC thought affecting aboveground biomass. Note that BRT model also is mentioned in the discussion of topography effects, as it provided information about potential paths through which topography would be exerting its effects on SOC. However, as both referees have the similar questions about the BTR model, we would be opened to move it to supplementary materials or even suppress it you find that our explanations and the information provided by the model are not relevant enough for this manuscript. L. 487 SOC decrease with increase of slope Change done. (L 511) L.489 Not clear. . . To clarify this sentence, we changed it as follows: "In addition, high TSIS values compensated SOC stocks decrease with increase of slope, which could be due to reduced carbon inputs and increased carbon losses induced by steeper slopes" (L 511 - 513) L.491 What I see is that SOC stocks are lower under low intensity of grazing for low values of TSIS. . . "We changed the sentence as follows: "Increases in grazing pressure elevated SOC stocks under low TSIS values (Fig 3D)." (L 514) L.494-499 It is not really clear. We changed some sentences in this paragraph to make it clear: "Increases in grazing pressure elevated SOC stocks under low TSIS values (Fig 3D). This would be a surprising result according to recent meta-analyses, which conclude that grazing has a commonly decreasing effect on SOC (Abdalla et al., 2018; Eze et al., 2018; Mcsherry and Ritchie, 2013). However these effects were strongly context-specific, depending on other factors like climate and soil type vegetation (Abdalla et al., 2018; Eze et al.,

2018; Mcsherry and Ritchie, 2013). Moreover, light and medium grazing intensities can increase SOC inputs by dung deposition and promoting aboveground and root biomass production (Franzluebbers et al., 2000; Zeng et al., 2015). Considering that in our natural grasslands all grazing intensities are relatively low (see methods), our medium and high stock rates may increase soil carbon inputs in low seasonality locations by enhancing aboveground and belowground productivity." (L 514 - 525) L.507 high soil water contents? To clarify, we changed the sentence as follows: "In our study, high soil water contents caused by high MAP may inhibit decomposition if a shortage of oxygen supply occurs (Xu et al., 2016b)." (L 533 - 534) L.525 "which might be explained by" instead of "which is an indicator" Change done.

Please also note the supplement to this comment:
https://www.biogeosciences-discuss.net/bg-2020-63/bg-2020-63-AC1-supplement.pdf

---

## Author Response (AR1)

**Author´s response**

Dear editor:

In the following lines you can find the answers to your comments. As we did a new deep revision of the text, we also have added an updated version of the answers to the reviewers.

Regards,

The Authors

**Editor comments:**

Comments to the Author:

Dear authors,

Thanks a lot for responding to the reviewer comments. Both reviewer found that this manuscript presents an interesting data set and makes a significant contribution in exploring factors of SOC storage in grasslands. However, both reviewer noted that there are several formal aspects, which can be clarified as outlined in your responses. However, the overall writing has to be improved considerably.

We really appreciate the comments and suggestions the reviewers and you have done. The manuscript has definitively improved a lot. We have done an especial effort in improving the overall writtling.

In addition to the reviewer, I have the following comments:

1. In the Abstract make clear on how many sites your study is based on and to which depth samples have been taken.

Changes done. (L 87)

2. Take care that all abbreviations are explained properly. E.G. BRT might be known by statisticians but not by all soil scientists.

We checked the remaining abbreviations in the text, and under our view they are well explained. In the case of BRT models, we added the following information.

"We applied two different modelling procedures: Boosted Regression Trees (BRT) and General Linear Models (GLM). BRT is an automatic technique that combines insights from traditional statistical modelling and machine learning traditions (Elith et al., 2008). GLM allowed us to design a hypothesis-based modelling procedure, ensuring that only effects with biological meaning where included; whereas BRT provided information about the data that could be neglected, if only a GLM approach was followed." (L331-336)

3. Tables. Explain all abbreviations (especially the not well known ones, e.g. TSIS). Tables and Figures should be self-explanatory

We added abbreviation explaining in all table and figure captions, including supplementary material.

4. Final model. Is it meaningful to include soil N? More than 90% of N are bound to soil organic matter and consequently N must correlate with C, at least their concentrations. Due to this inherent relationship, I would argue that N is a controlling factor. In the model you have used to explain stocks (whose estimate includes bulk density), but you have to provide the relation between C and N concentrations and then the reader may judge how meaningful soil N is.

We agree with this comment at 90%, and we find it useful and interesting. We would say that although it is true that SOC and soil N must be correlated as they constitute soil organic matter, due to the wide range of conditions and the randomized sampling design of the PASTUS database, raw correlation between soil N and SOC was something discrete ($r = 0.88$; p-value = 0.001; $R^2 = 0.09$) when comparing with other studies (i.e. Yan *et al.* 2020). What our models proposes is that soil N modulates SOC response to certain drivers (grazing management and NDF). In other words, grazing management and NDF effects on SOC differ depending on soil N conditions. So soil N would not be completely a controlling factor, although is not as the rest of SOC predictors because of the reasons you exposed.

To explain this, we modified the corresponding paragraph in the discussion section as follows:

"A positive relationship between SOC and soil N was also expected, since most of the soil N is in combined form with organic matter (Cambardella and Elliott, 1994). However, in this case, due to the wide range of conditions and the randomized sampling design of the PASTUS database, the raw correlation between soil N and SOC was somehow discrete ($r = 0.297$; p-value = 0.001; $R2 = 0.088$), in comparison to other studies (i.e. Yan et al. 2020). However, the novelty revealed by our model is that soil N could modulate the effects of certain SOC drivers, including livestock type and herbage NDF." (L 629-635)

Could you so provide support that the final model indeed allows that many factors (e.g. by showing AIC)?

We consider that the number of samples (128) is high enough for allowing the number of factors in the Combined Model. Additionally, as we explained in the manuscript, we did not use AIC in the modelling procedure, but the corrected version AICc, which penalizes much more the inclusion of additional terms, hence ensuring that new included factors contribute with really significant information (Burnham & Anderson 2002; Burnham *et al.* 2011). As far as we know, AIC and AICc of the whole model do not give information about if the number of factors is appropriate, but about the information explained by the model in relation to other models. Anyway, AICc of the Combined model was -14.16466. Additionally, below this lines you can find a table showing AICc changes (ΔAICc) in the Combined model when removing each main variable or interaction. Results for main variable effects show changes when removing the main effect and all the interactions using that variable. F test and p-value columns show the anova test between the Combined model and the model resulting of removing that variable or interaction. In this table you can appreciate how excluding any term implies an important information loss (ΔAICc > 2; (Burnham & Anderson 2002; Burnham *et al.* 2011)); and that those models are significantly different according to anova test between models. We decided not to include this table because we find it redundant with Table 3, which provides significance test for each model term and also the coefficients of the model. We could include them as supplementary material or even in the main document if you consider it necessary.

| Variable or interaction term | ΔAICc | F test | p-value | |
|---|---|---|---|---|
| TSIS | 26.58 | 9.26 | 0.000 | *** |
| Slope | 4.89 | 4.65 | 0.012 | * |
| MAP | 3.97 | 5.93 | 0.017 | * |
| Soil N | 62.15 | 20.87 | 0.000 | *** |
| Log(C/N) | 66.05 | 77.03 | 0.000 | *** |
| Grazing species | 10.86 | 4.28 | 0.001 | *** |
| Grazing intensity | 10.52 | 4.98 | 0.001 | ** |
| NDF | 12.30 | 8.14 | 0.001 | *** |
| ADL/NH | 10.65 | 5.79 | 0.001 | ** |
| Soil N x Grazing management | 15.95 | 9.94 | 0.000 | *** |
| TSIS x Slope | 6.31 | 8.03 | 0.006 | ** |
| Soil N x NDF | 10.92 | 12.28 | 0.001 | *** |
| TSIS x Grazing species | 5.29 | 4.83 | 0.010 | ** |
| Grazing species x ADL/NH | 3.78 | 4.14 | 0.019 | * |

5. How did you consider that factors are autocorrelated e.g. pH and Mg?

We are not sure of understanding if this question refers to spatial autocorrelation or to correlations between variables. Anyway, as our response variable was not spatially autocorrelated according to Morans´s I test (z-score = -1; p-value = 0.3), we do not consider that the fact that explanatory variables were autocorrelated or not could have any noticeable impact for modelling procedure. In other words, spatial autocorrelation could be a problem if it were presented by the response variable (soil organic carbon), not the explicative variables. Additionally, the randomnized sampling design of the PASTUS database probably reduced the possibility of having spatially autocorrelated variables.

Concerning the correlations between explanatory variables, the randomized sampling design of the PASTUS database and the GLM modelling procedure were designed to reduce and control that issue. However, GLM procedure did not selected Mg or pH not because they were highly correlated with other variables, but because the effect they could have on soil organic carbon were surely included in other variables with more explanatory power. As BRT is a modelling procedure that includes explanatory variables regardless of the information provided by other variables, we can detect that Mg and pH had some effects on soil organic carbon. To identify which concrete variable(s) produce pH and Mg displacement in the GLM procedure would imply an exhaustive exercise of forward-backward modelling.

In the text, this is explained as follows:

"Furthermore, BRT model provided some valuable information, identifying some relevant SOC drivers which were discarded during the GML modelling, like aboveground biomass, or soil silt and K (Fig. 2 and S8). The effects of those drivers were probably masked by the effects of other variables in our linear models (Yang et al., 2009), indicating that these factors were presumably pathways through which other variables drove SOC (de Vries et al., 2012). These variables, identified by BRT and discarded by GLM, should be considered as potential SOC drivers in further studies, particularly when more detailed and difficult to obtain biochemical variables, present in our database, are not available." (L 473-476)

We made several changes in the explanation of this point in the methods and discussion, and we truly believe now is clearer and more understandable than before.

6. Table 1. Is it really meaningful to show what has been measured in which study but not showing the actual outcome of these studies? For instance SOC stocks, main controlling factors, etc.

The aim of including that table was to highlight the wide variety of climate conditions that the PASTUS database contains and the different soil organic carbon drivers that includes. In a previous study about species richness done with dese database (Rodríguez et al. 2018), we get some criticism from certain journals because the argued that the range of study was "too regional". This time, we really appreciate the recognition that both referees had done to the effort of compiling this amount of data.

We added some of the information to the table. In the caption, we specified which studies considered other response variables instead of soil organic carbon (total carbon stocks, soil organic carbon concentration etc.). However, adding more information is not an easy task. Soil organic carbon ranges are difficult to summarize because each study considers different soil depths and there would be necessary an intensive work to get comparable values. We included the types of explanatory variables included in each study, but specific variables and effects and so heterogeneous that are difficult to include in a table, and could lead to the reader to misleading conclusions about these papers.

7. Figure 2: Clarify that the relative importance is related to explained variance.

The way that relative importance of explanatory variables is not exactly related with the concept of explained variance. According to Elith et al. (Elith et al. 2008):

"The measures are based on the number of times a variable is selected for splitting, weighted by the squared improvement to the model as a result of each split, and averaged over all trees (Friedman & Meulman 2003). The relative influence (or contribution) of each variable is scaled so that the sum adds to 100, with higher numbers indicating stronger influence on the response."

If you consider it appropriate, we can include this information on the methods, although we thing it could be a little bit excessive. In the caption of Fig. 2 we added: "Higher numbers indicate stronger influence on SOC stocks (Elith et al., 2008)."

8. The explanation of interaction seems very speculative.

We revised once again both the manuscript and the literature for making our discussion less speculative. Finding experimental studies to support our results is a difficult task because experimental studies assessing interactions between SOC drivers are not as frequent as one could expect. This is a problem that affects not only to SOC, but to soil properties in overall. For instance Rillig *et al.* (2019) revised 1228 experimental studies about global change drivers of diverse soil properties, published between the years 1957 and 2017, and they found that 80% looked just at the effects of one single factor and 19% at the interaction of two factors. This pattern did not changed over the time, so this shortage of experimental studies about interactions is also manifested in current publications. To make things less favourable, in the mentioned revision fertilization was one of the tested factors in more than half of the experiments assessing interaction effects, which suggest that natural or at least extensively managed grassland conditions are poorly represented in experimental studies. Being conscious of these limitations, we are convinced that of work will provide highly valuable information. Interaction experiments are expensive, since they require high sample sizes. Patterns observed in studies like ours can be useful to propose hypothesis to test instead of raising hypothesis blindly.

We added some of this information in the discussion section:

"Those results must be interpreted cautiously, because they are based on observational data, but can contribute to generate testable hypotheses for later studies about some complex and untested relationships between SOC and its drivers. Interaction experiments concerning soil properties are expensive and rare in the literature (Rillig et al., 2019)." (L 621-625)

However, if after reading our revised manuscript you find that some parts of the discussion still need an improvement, we would be pleased of listening your suggestions. We find that our manuscript has improved a lot with this revision.

9. Language should be carefully checked. It can be smoothed at many places.

We have intensively checked language. However no one of the authors is an English native speaker. Our institution has a language revision service, but tis August was closed because of COVID. If you still find that language has to be improved after this revision, we would be able to send it to that service as of September.

Overall, the manuscript entitled 'Interactions between biogeochemical and management factors explain soil organic carbon in Pyrenean grasslands' would have potential to be of great interest for the readers of Biogeosciences Journal. It provides interesting results on the effect of different drivers on soil carbon stocks in Pyrenean grasslands. However, I have noticed some important points that need to be addressed before this manuscript can be considered for publication.

Concerning the abstract, I think that the scope and objectives of the study need to be better defined. After reading it, we do not have a clear idea of what factors have been tested. I have the same feeling after reading the introduction. Overall, we understand that there are many factors which can influence soil C stocks at different scales, but it is difficult to understand what are the real objectives of the study. Is the objective to determine which factors influence the most the soil C stocks, is this analysis done for different scales?

We have revised the abstract and the introduction sections, following your specific comments. Under our view, the scope and objectives are now more understandable. In a nutshell, the scope is to study the relative effects, including interaction effects, of geophysical and biochemical SOC drivers, and also to pinpoint how grazing management regulates the effects of other SOC controls.

In the material and methods section, the main issue that I noticed concerns the statistical approach. It is not clear for me why two separate approaches were done. It adds a certain complexity to the article and it needs to be better presented according to the objectives for each approach. Are both the approaches really relevant for the paper? The links between the objectives and the chosen modelling approach needs to be better defined. Also, concerning the calculation of soil C stocks, it would have been appropriate to correct soil C stocks according to the equivalent soil mass approach to account for possible differences in bulk density values (Ellert and Bettany, 1995; Ellert et al., 2008).

We explained in the specific comments why we think both statistical approaches are complementary and important. We revised the manuscript to emphasize and make clear this point. However, if our arguments neither convince nor the editors nor the referees, we are open to put the BTR model in supplementary material or even suppress it completely. Note that although it has been argued that the usefulness of using both approaches was not clear, referee 1 made several specific questions about the differences between their results, precisely about the points we consider interesting. We also commented the point about fixed mass approach for calculating SOC stocks in the corresponding specific comment.

Concerning results and discussion, even if the ideas are, overall, well supported by relevant references and the limits are underlined, I think that the organization will be improved after the clarification of the objectives and the corresponding analyses. Also I noticed repetitions of results in the 'results' section and in the

'discussion' section so I would suggest to group all the results and discussion in one section if the journal guidelines allow it.

We think separate sections for results and discussion are important, since this is useful for separating the raw statistical results from results discussion and interpretations. We truly believe that the manuscript is going to be easier to read and understand if we maintain this structure. The statistical methods presented here could seem complex, and reading the results separately could help to their understanding since is the shortest and simplest section. Anyway, we followed your advice and we revised the manuscript to make it less repetitive. Your specific comments where greatly valuable for this task. The most important change is we suppressed the first paragraph in the discussion section, which was actually a summary of the result section. We also revised the paragraph about the modeling procedure, and we believe now is more clear.

We think the rest of the subsection titles in the discussion were useful to structure the text. Under our view, every sub-section was justified. However, we grouped subsections 2-4 (Geophysical, biochemical and grazing management factors driving SOC stocks) as both referees asked us to reduce the number of sections. The idea is that first section gives an idea of the right way of interpreting the models. The second section answers the questions formulated at the end of the introduction; 1: "what are the relative and interaction effects of the geophysical and biochemical SOC controls?" and 2: "How grazing management regulate the effects of other SOC drivers? Finally, we separated and revised the conclusion section following the indications of referee 2.

Of course, if after this revision, the referees and the editor consider that results and discussion section must be combined, we could do it without a problem.

Finally, it would be important that the manuscript be reviewed for the English. Some corrections might be necessary.

We revised the English.

In the next paragraph, I developed some detailed comments that will help the authors to improve the manuscript.

L 53-54 "at small spatial scales" instead of "at detailed spatial scales".

Change done. (L 59)

L 56-57 I am not sure that it is a good reason to do a study... What is the objective of the study by using this set of data?

To clarify this, we rewrote this sentence as follows:

"Taking advantage of the high variety of environmental heterogeneity in the Pyrenees, we built a dataset (n = 128) that comprises a wide range of environmental and management conditions. This was used to understand the relationship between SOC stocks and their drivers considering multiple environments." (L 61)

L 58 Do the authors have an explicative purpose or a predictive purpose? That is not clear for me, as they also use the 'predictors' term.

The study has an explicative purpose. We have changed "predictors" by "drivers" or "factors" in all the text to avoid misinterpretations.

L 59 This factor should be better defined.

We specified it in the following way:

"We found that temperature seasonality (difference between mean summer temperature and mean annual temperature; TSIS) was the most important geophysical driver of SOC in our study." (L 65-66)

L 65 I think that the coma is not necessary.

The comma was removed. (L 71)

L. 95-96 I think that these variables should be better described. Also "be" should be removed.

These factors are not studied or they are not factors with a relevant impact in other studies?

This phrase was rewritten as follows, to clarify that these factors were not even considered in these previous studies and the meaning of climate seasonality:

"However, climate annual variations, represented by seasonality variables, are be commonly neglected when considering possible SOC drivers affecting SOC in broad-scale models, in spite of being some important factors for plant primary production or enzymatic activity of soil microorganisms." (L 104 - 108)

L.112 Same question than earlier: are they omitted because they do not impact the SOC stocks?

The phrase was rewritten as follows:

"However, these variables are commonly omitted as possible drivers of SOC in the broad-scale studies, especially in those studies focusing on predictive rather than explicative models" (L 122 - 125)

L. 147 "focusing" instead of "focus"

Change done. (L 123)

L. 116 Overall, for the whole manuscript, the authors need to specify if it is SOC stock or concentration.

It is SOC stocks. That is now specified in multiple parts of the text.

L.127 What type of management do you consider?

In our case (natural grasslands), we consider livestock management. However, according to the cited article (Wiesmeier et al., 2019), management effects on grassland SOC in general are poorly understood. We have rephrased the sentence:

"In addition to these factors, livestock management effects on grassland SOC is …" to clarify that in this paper we only refer to livestock management, which is from far the main management done in natural grasslands." (L 138).

L. 136 And what was their conclusion in regards of your objectives?

The conclusion in regards of our objectives was that grazing must be considered as a variable that can interact with many variables at multiple scales (as it is represented in Fig. 1). We reordered this paragraph and completed this particular sentence to clarify this point:

"It is known that herbivores can affect SOC through different paths, such as regulating the quantity and quality of organic matter returned to soil (Bardgett and Wardle, 2003), or affecting soil respiration and nutrients by animal trampling or soil microbiota alteration (Lu et al., 2017). Several studies confirmed the interaction between grazing and other SOC drivers at diverse scales (Abdalla et al., 2018; Eze et al., 2018; Lu et al., 2015, 2017; Zhou et al., 2017). Hence, grazing management may be considered a SOC driver with effects at multiple levels of the driver hierarchy (Fig. 1), both affecting other SOC drivers and interacting with them. However, most of the studies investigating grazing effects on SOC focus on grazing intensity, in spite of evidence pointing to a greater role of grazer species in determining vegetation and SOC (Chang et al., 2018; Sebastia et al., 2008)." (L 139-151)

 L. 140 Among which drivers? There are many factors that can interact or be correlated together. We need to know which drivers will be tested. The authors should be clearer on the objectives of this study.

We modified this paragraph as follows:

"In this study, our goal was to identify the main drivers of SOC stocks and their interactions on Pyrenean mountain grasslands. For this purpose, we considered a wide set of regional, landscape, soil geophysical and biochemical and herbage quality factors, together with grazing management factors. Mountain grasslands comprise a wide range of all these conditions which make carbon stocks highly variable (Garcia-Pausas et al., 2007, 2017). For this reason data analyzed here comprise a wide range of environmental conditions, comparable to studies on SOC developed at continental or even worldwide scales (Table 1). Additionally, we consider an exceptionally broad compilation of drivers (Table 1). To deal with correlations and interactions between SOC drivers, we developed an exhaustive modelling approach based on the controls over function hypothesis (de Vries et al., 2012)." (L 152 - 161)

L. 141 To asses

This sentence was changed and this word does not appear.

151-153 Do the authors want to study the effects of various factors, their links between them, the importance of the factors...?

We rewritten the questions as follows to put that point clear:

"1) What are the relative and interaction effects of the geophysical and biochemical SOC controls? 2) How grazing management regulate the effects of other SOC drivers?" (L 170)

L.175 grazer type instead of grazing management

Change done

L 189-190 Are the soil samples from the 4 quadrats composited to form one soil sample per depth for each grassland patch?

Yes, they are. Following the advice of referee 2, we made many clarifications about sampling design, including a new supplementary figure (Fig. S2).

L. 192-193 I think this paragraph should appear before...

We appreciate this comment, and we also recon that this paragraph could appear at the beginning of the methods section. However, we still find clearer to explain first how the sampling was performed and second how the samples were processed in order to get the environmental variables.

L. 194 There should be a coma between landscape and livestock

Change done (L 216)

L. 199 But you don't speak of mean summer temperature before...

We added MST where climate variables were introduced, in the second paragraph of the section 2.2: Regional and Landscape environmental drivers:

"Regional variables included climate variables and bedrock. Climate variables were determined from Worldclim 2.0 (Fick and Hijmans, 2017). We selected Mean Annual Temperature (MAT), Mean Summer Temperature (MST), Mean Annual Precipitation (MAP) and Mean Summer Precipitation (MSP)." (L 218 - 221)

L. 200-201 How did you appreciate that? We need to have more details on this factor.

During preliminary modelling exercises, two climatic variables appeared repeatedly in all tested models, always significantly contributing to the models with coefficients of similar magnitude but opposite signs: mean annual precipitation and mean summer temperature. In those initial models, even when those two variables were included in interaction with other drivers, this pattern was maintained; that is, the interactions were of opposite sign and the coefficients of the interactions were similar in magnitude. In this way, this is how the TSIS index initially emerged. This index has been found a significant driver of different variables of interest in the PASTUS database, including SOC and plat diversity (Rodríguez et al. 2018).

We added the following explanation:

"The difference between mean annual and mean summer temperature emerged as a relevant explanatory factor of soil organic carbon stocks during previous modelling efforts by one of the co-authors (M-T. Sebastià). Later attempts to improve models by substituting this variable with other temperature indices from climatic databases (Fick and Hijmans, 2017) showed that, for the PASTUS database, this variable provided higher explanatory power than other temperature seasonality indices. Thus, we decided to keep it and here we name it Temperature Seasonality Index of Sebastià (TSIS from now on)" (L 2221-228).

L. 218 For each patch considered?

Those grasslands are usually managed communally, and the livestock type and units are based on the number of animals, and type, sent to graze a given area during the grazing season. The unit area is usually related to the municipalities, although this situation might change a little depending on the mountain range. Grazing in the high-altitude grasslands in the Pyrenees is usually free-range.

L. 229 For determination of bulk density?

Yes. We modified the sentence as follows to clarify this point:

"To obtain bulk density, we air-dried and weighed the soil samples: we then sieved each sample to 2 mm to separate stones and gravels from the fine earth fraction. " (L 242)

L. 233-234 This sentence is not clear.

We rephrased this sentence to clarify it:

"We combined 0-10 and 10-20 cm values for obtaining the whole top 20 cm soil layer." (L 247 - 248)

L. 243 It should have been important to correct soil C stocks according to the equivalent soil mass approach.

We decided to use a fixed depth approach for calculating SOC stocks due to the following reasons. First, the main advantage of fixed mass approaches is that they account and correct differences in bulk density due to temporal changes or when comparing different land uses (Haden et al. 2020). We do not consider variations in time, and neither have contrasting management regimes, as mentioned in the title of Ellert & Bettany´s paper (1995). We highlight that in our work samples came from natural mountain grasslands, where grazing intensity is always low to moderate, and moreover, herbivore presence is seasonal. Therefore, we do not expect important changes in bulk density due to land use. Second, we always used the same methods in our samplings (so we could not take advantage of fixed mass approaches for correcting biases due to different probe diameters, as suggested by Sharma et al. (2020). Finally, fixed mass approaches often have more technical difficulties than fixed depth measures even in the most modern procedures (Haden *et al.* 2020). On the other hand, Rovira et al. (2015) proposed a fixed mass approach which, as expected, was found to deal properly with bulk density changes but not with stoniness differences. We did not find any other reference dealing with this point.

To clarify this point, we added the following lines to the text:

"Soil organic carbon (SOC) stocks in the upper 20 cm soil layer were then estimated taking into account the organic C concentration in the sample and its bulk density, and subtracting the coarse particle (> 2 mm) content, following García-Pausas et al. (2007). Despite recent studies suggesting that fixed mass SOC stocks estimates are preferable to fixed depth methods because they would be more robust to temporal and land use changes in bulk density (Ellert & Bettany 1995), we chose a fixed depth method for measuring SOC stocks. This decision was based on the fact that our work samples came from natural mountain grasslands, where grazing intensity is always low to moderate, and moreover, herbivore presence is seasonal. Therefore, we do not expect important changes in bulk density due to land use. Additionally fixed mass approaches  presented the disadvantages of implying more technical difficulties than fixed depth measures, even in the most modern procedures (Haden et al. 2020), and could not deal well with differences in stoniness." (L 259 - 271)

L. 249 What was the vegetation: grassland species etc.

We added the following paragraph to provide that information:

"Almost all of the plant species in the grasslands from the PASTUS database are perennial (Sebastià, 2004), and plant biodiversity is highly heterogeneous as are the environmental conditions (Rodríguez et al., 2018)." (L 185-187)

As this is not bromatological information we added this paragraph in lines 209 - 211 where describing the sample site conditions.

L. 267 The size of the police is not the same for all this paragraph. Does this paragraph of NIRS analysis refer to the analyses presented in the previous paragraph? It is not clear.

We changed some sentences in these paragraphs so now the relationship between these two methods is clear. Basically, bromatological analysis were done for training NIRS models and getting the remaining values using NIRS spectrum. (L 272)

L. 293 Among which variables?

Among SOC and all the considered drivers. To clarify, we modified the sentence as follows:

"Including all SOC potential drivers, we fitted a model with BRT to identify the most important ones affecting SOC." (L 340)

L.301 "Firstly" instead of "First"

Change done. (L 348)

L. 306-307 What is this new set of variables?

This is standard procedure, according to (Elith *et al.* 2008). As it is explained, it refers to the variables that improve BTR model performance, the model set showed in Fig. 2.

L. 314-316 Why choosing these two models, on which hypothesis did you decide these two groups?

The geophysical variables are those commonly used in the literature, and are the first source of variation according to the hierarchy of controls over function hypothesis (Manning et al., 2015). Choosing these two models allows us to discuss the effects of geophysical variables on SOC without deleting some effects because of the inclusion of other variables (especially soil nutrients) whose effects may include those of geophysical variables, because geophysical variables could act trough other variables at smaller spatial scales (in this case, the biochemical variables). We consider Geophysical Model is interesting for discussion, since it allow comparisons with previous literature. Additionally, we reported which terms of the Geophysical Model were substituted by the biochemical variables, which suggests that those effects could affect SOC trough biochemical variables, while the other effects probably acted trough other mechanisms too. Finally, we believe that Geophysical Model has interest for future studies aiming to predict SOC in similar environmental conditions. As we mentioned before, these studies usually use what we call here geophysical variables, because they are easy and cheap to measure or obtain (Manning et al., 2015). We modified the referred sentence as follows, to emphasize some of these points:

"We built two models (Fig. S5), one model based only on geophysical drivers and grazing management (Geophysical Model), and another model including, in addition to the former drivers, the biochemical drivers: soil nutrients and herbage quality (Combined Model). With this approach we aimed to avoid ignoring significant effects of the geophysical variables, the original source of variation of SOC stocks according to the hierarchy of controls over function hypothesis (Manning et al., 2015), by masking them with the inclusion of biochemical drivers." (L 360 - 373).

We also added the following modifications in the "Geophysical drivers driving SOC stocks" subsection of the discussion section to a better explanation of the usefulness of the Geophysical model:

"Considering the difficulties of modelling SOC in a widely heterogeneous mountain environment (Garcia-Pausas et al., 2017), the Geophysical Model provided important information about broad-scale and topographic SOC drivers in the Pyrenees. This information could be useful not only for a better understanding of SOC patterns in mountain grasslands, but also for future modelling studies aiming to predict SOC, since geophysical variables are easier and less expensive to acquire and measure compared to biochemical variables (Manning et al., 2015)." (L 484 – 490)

L. 316-320 Maybe it should be more appropriate in the introduction...

We think this is appropriate for the methods as it contributes to the understanding of the modeling procedure. However, we modified the last paragraph of the introduction, to specify these aspects too. In overall we believe that this important paragraph has been widely improved thanks to your comments and suggestions. This text is now as follows:

"In this study, our goal was to identify the main drivers of SOC stocks and their interactions in Pyrenean mountain grasslands. For this purpose, we considered a wide set of regional, landscape, soil geophysical and biochemical, and herbage quality factors, together with grazing management factors. Mountain grasslands comprise a wide range of all these conditions, which make carbon stocks highly variable (Garcia-Pausas et al., 2007, 2017). For this reason, data analysed here include a wide range of environmental conditions, comparable to studies on SOC developed at continental or even worldwide scales (Table 1). Additionally, we considered an exceptionally broad compilation of drivers (Table 1). To deal with correlations and interactions between SOC drivers, we developed an exhaustive modelling approach based on the controls over function hypothesis (de Vries et al., 2012). To facilitate the formulation of our specific questions to answer in this study, we classified SOC drivers into three main groups (Fig. 1): i) geophysical factors, which include regional and landscape factors and are supposed to be the first sources of variation, ii) biochemical factors, which include soil nutrients and herbage factors and could be conditioned by geophysical factors, and iii) grazing management factors, which could affect SOC through multiple interactions with the rest of the variables at multiple scales. In particular, the specific questions of this study are 1) What are the relative and interaction effects of the geophysical and biochemical SOC controls? 2) How does grazing management regulate the effects of other SOC drivers?" (L152 - 172)

L.374 Why there are not all the predictors described in the introduction in this model? Grazing management for example?

Because they were discarded in the BTR modelling procedure. Note that three-based methods can have difficulties in modeling some functions (Elith *et al.* 2008). We answered about why using both methods three comments below.

L.381 Why these two variables are not selected in the model?

This point was discussed in the "Considerations about the modelling procedure" subsection inside the discussion section. "Furthermore, BRT model provided some valuable information, identifying some relevant SOC drivers which were discarded during the GML modelling, like aboveground biomass, or soil silt and K (Fig. 2 and S8). The effects of those drivers were probably masked by the effects of other variables in our linear models (Yang et al., 2009), indicating that these factors were presumably pathways through which other variables drove SOC (de Vries et al., 2012)." (L 473-478).

Basically, multiple predictor variables can not only be correlated but also have true cause-effect relationships between them (i.e. precipitation and aboveground biomass), what means that in a linear model, some drivers could be discarded not because they have no effects on the response variable, but because their effects were already included in other variable. In other words, some variables, like aboveground biomass, soil K or silt were not included in the linear models probably because they were correlated with other drivers which were included in the models. The advantage of including BTR analysis is that we could detect some of these variables.

L.411 Some repetition from the results section...

We deleted this paragraph as it is repetitive.

L.444-447 I wonder if the BRT model is really relevant for the manuscript. . . Also, Are you sure it is table S3???

It is table S5 (change done). To summarize, the BTR model is relevant insofar it provides information about the effects of the variables not included in the linear models due to correlation. Is this information relevant enough for the manuscript? We think it is. For instance, if BTR model were not included, one question a referee or regular reader would ask would be: "How can you explain that aboveground biomass was not included in your models?" "Does it means that aboveground biomass had no effects on SOC?" The answer is that aboveground biomass had effects on SOC, but in the GLMs these effects were masked by other variables which explain more variation than aboveground biomass, and probably affect SOC thought affecting aboveground biomass. Note that BRT model also is mentioned in the discussion of topography effects, as it provided information about potential paths through which topography would be exerting its effects on SOC. However, as both referees have the similar questions about the BTR model, we would be opened to move it to supplementary materials or even suppress it you find that our explanations and the information provided by the model are not relevant enough for this manuscript.

L. 487 SOC decrease with increase of slope

Change done. (L 522)

L.489 Not clear. . .

To clarify this sentence, we changed it as follows:

"In addition, SOC stocks decreased with increase of slope, which may be attributed to reduced carbon inputs and increased carbon losses induced by steeper slopes" (L 522 - 523)

L.491 What I see is that SOC stocks are lower under low intensity of grazing for low values of TSIS. . .

We changed the sentence as follows:

"At low TSIS values, SOC stocks increased under moderate to high grazing pressure; this effect disappeared as TSIS values increased (Fig. 3D)" (L 527-528)

L.494-499 It is not really clear.

We changed some sentences in this paragraph to make it clear:

"At low TSIS values, SOC stocks increased under moderate to high grazing pressure; this effect disappeared as TSIS values increased (Fig. 3D). Recent meta-analyses concluded that intensive grazing commonly has decreasing effects on SOC (Abdalla et al., 2018; Eze et al., 2018; Mcsherry and Ritchie, 2013). However, these effects were strongly context-specific, depending on other factors including climate and soil type vegetation (Abdalla et al., 2018; Eze et al., 2018; Mcsherry and Ritchie, 2013). Moreover, moderate grazing intensities can increase SOC inputs by dung deposition, and aboveground and root biomass production (Franzluebbers et al., 2000; Zeng et al., 2015). In our study, grazing intensity was relatively moderate (see methods), therefore in our study increasing stocking rates may increase soil carbon inputs in moderate seasonality locations by enhancing aboveground and belowground productivity." (L 527 - 537)

L.507 high soil water contents?

To clarify, we changed the sentence as follows:

"high MAP may inhibit decomposition if a shortage of oxygen supply occurs (Xu et al., 2016b)." (L 541)

L.525 "which might be explained by" instead of "which is an indicator"

Change done.

Referee 2

General comments

The manuscript aims to understand how environmental and management factors affect SOC in mountain grasslands. And fitted a set of models with explicative purposes using data that comprise a wide range of environmental and management conditions to find the most important driver of grassland SOC. The authors are to be commended on the framing of an interesting study, the collection of a reasonable set of ancillary environment and management data and soil data in what appears to be good quality piece of research. The workload of this article is very huge.

However, too many sections and repetitive statements in this article. Be better structured and more concise to attract readers.

Please, see our answers to referee 1 about the modifications in the text structure.

Deep discussion and comparison of your work is needed in an international context. In discussion section, some discussion on the mechanism of environmental and management factors should be added.

We would really appreciate it if you could specify more about which mechanisms need more discussion. Referee 1 found that discussion section was "overall, well supported by relevant references and the limits are underlined". We recon you have a point concerning biochemical or management species effects on SOC: the mechanisms are not widely explained but, as we explain in the text, that is a difficult task since there are few publications addressing these issues. We revised the published works from this manuscript was sent to Biogeosciences until now and, under our view, no remarkable novelties have appeared in these topics. However, we found a bibliometric study which concluded that interaction experiments concerning soil properties are expensive and rare in the literature (Rillig et al., 2019). We would appreciate it if suggestions about ideas or publications we could omit were made in order to improve the manuscript.

I suggest you add a conclusion section, a concise and clear conclusion will make your article more eye-catching and let readers understand the conclusion of this article more quickly and easily.

We separated the conclusions from the discussion section, and we changed that paragraph to make it as much clear and concise as possible, focusing on the main contributions of our manuscript to scientific knowledge.

As the manuscript contains some uncertainties in description of the methods, results, and English writing, I suggest a moderate revision necessary before it can be acceptable for publication in this journal.

We corrected the uncertainties in the text. The specific comments of both reviewers were really helpful and we really appreciate them.

Specific comments

Line 75 "Soil organic carbon plays key roles in the terrestrial ecosystems." It sounds strange.

We rephrased this sentence as follows:

"Soil organic carbon (SOC) is crucial for the functioning of terrestrial ecosystems." (L 84)

Line 179 At least one to two replicates of each patch type were sampled. What are the types of the patch?

To clarify this point, we rephrase this sentence as follows:

"Grassland patches were then listed by type and arranged within each list randomly to determine sampling priority. At least one to two replicates of each patch type defined by the stratification variables were sampled." (L 199 - 201)

Line 155 Not clear sampling design description. Showing a figure with sampling design would be helpful. Add a schematic of experimental design to make it clearer.

We added figure S2, which illustrates sampling design.

Line 192 The abbreviation for soil organic carbon had appeared in line 75, here only need to write SOC.

Change done. (L 214)

Line 193 There are 30 variables written in table S1, but here you have written 29 independent environmental variables. Are the two management variables belong to environmental variables? Please check these numbers.

Change done. (L 215)

Line 194 These variables were grouped into Regional, landscape, livestock management, soil nutrient stocks, and herbage variables? If so, replace ":" with ",".

Change done. (L 216)

Line 201 MTS?

M-T. Sebastià. We changed this to make it clear. (L 223)

Line 220 Here used livestock stocking rates which measured as livestock units ha-1 to determine grazing intensity. But the feed intake of different types of livestock is different. For example, the intake of cattle is higher than that of sheep. So, can't simply use the livestock units/ha-1 as livestock stocking rates, you need use standard livestock unit.

We used a standard transformation index where 8 small ruminants correspond to 1 big ruminant. This is standard and provided by the Catalan Government for the region.

Line 314 Geophysical model based on geophysical predictors and grazing management? There haven't grazing management in Figure S4.

Now Figure S4 has grazing management.

Line 371 Authors need to better describe statistics of SOC.

We added some information about the statistics of SOC. However, we do not know what more to add apart of basic descriptive statistics we already show. We will really appreciate it if you could specify what statistics you miss in this part of the text. (L 420 - 423)

Line 375 Generally, a part of the sample is used for modeling, and the other part is used for validation. Please describe clearly in here and in Line 279.

Concerning the line 375 (now 425) (BRT model) we did not validate the whole model with a fraction of the dataset, because our BTR model was fitted by cross validation (CV; it is used to select the number of trees with the best performance). Note that according to Elith et al. (2008), results of this cross validation procedure are often very similar to those obtained with independent datasets. Additionally, note that each tree was actually fitted with 66% of the data (out of the bag fraction parameter), so our procedure properly dealt with stochasticity too. All these are standard methods explained by Elith et al. (2008), so we prefer just to refer to this publication instead of extending our methods section, and to focusing on other parts of the statistical procedure that need to be clearer as possible. However, if you think that some of the standard aspects of the BRT procedure deserves to be explained in our manuscript, we will follow your advice.

We detailed the herbage-bromatological analysis (L 279 (Now 272) and so on). 130 samples where used for the validation of NIRS equations.

Line 379 Silt in here, loam in fig.2. Use consistent terminology of silt, loam, etc? Use one, Please!

Change done. Silt is now the only name used.

Line 382 Why TSIS was the most relevant selected climate predictor? In figure 6s, Soil C/N has a higher relative importance.

TSIS was the most relevant of the climate predictors (without considering other variable types). To clarify this point, we rephrased this sentence as follows:

"TSIS was the most relevant among the climate drivers considered." (L 432)

Line 383 Please confirm this sentence and the quoted figure. I didn't find TSIS in figure S5 and S6. In table s1, TSIS described as MST-MAT. In figure s8, MMT also described as MST-MAT Use consistent terminology of MMT, TSIS, etc? Use one, Please!

Change done. MMT is a previous nomenclature. TSIS is the proper one.

Line 381 Aboveground biomass and silt had a high relative contribution in the final BRT model obtained, why not selected them in the linear models?

This was also true for soil K and silt. This point was discussed in the "Considerations about the modelling procedure" subsection inside the discussion section. "Furthermore, BRT model provided some valuable information, identifying some relevant SOC drivers which were discarded during the GML modelling, like aboveground biomass, or soil silt and K (Fig. 2 and S8). The effects of those drivers were probably masked by the effects of other variables in our linear models (Yang et al., 2009), indicating that these factors were presumably pathways through which other variables drove SOC (de Vries et al., 2012). These variables, identified by BRT and discarded by GLM, should be considered as potential SOC drivers in further studies, particularly when more detailed and difficult to obtain biochemical variables, present in our database, are not available." (L 473 - 481)

Basically, as multiple predictor variables can not only be correlated but also have true cause-effect relationships between them (i.e. precipitation and aboveground biomass), what means that in a linear model, some drivers could be discarded not because they have no effects on the response variable, but because their effects were already included in other variable. In other words, some variables, like aboveground biomass, soil K or silt were not included in the linear models probably because they were correlated with other drivers which were included in the models. The advantage of including BTR analysis is that we could detect some of these variables. There is more about BTR models in some answers to referee 1.

Line 1121 Please add the fitting equation in figure 3 and 4. It is hard to distinguish which trend line belongs to which grazing species or grazing intensity. You can distinguish by color, or add the legend.

We changed all the plots to the main document to color plots, so lines and dots are more distinguishable than before. We also added the sentence "The estimates on Table 2-3 were those used to elaborate these plots." so the equation values can be easily found.

Line 25 in SUPPLEMENT Figure S1: points indicate sampling location, sampling location means the sample patches? Please add the legend of the points in this figure.

As we explained in the methods section, each sampling patch contains a sampling location, located in the middle of the grassland patch. Sampling location were added in the legend of this figure. As you suggested in some lines above, we added the figure S2 to clarify the sampling design, and the legend of the points in Fig. S1.

Line 39 in SUPPLEMENT There is no reference of Figure S3 in the text.

We added the reference in the "general linear models" subsection, in the material and methods section:

[revised manuscript text omitted]
; MAP: mean annual precipitation; MSP: mean summer precipitation; Slope: terrain slope; Aspect:; Sand: sand content; Loam: loam content; Clay: clay content; pH: soil pH; Soil N: soil nitrogen; Soil P: soil phosphorus; Soil C/N: soil carbon to nitrogen ratio; Soil Mg: soil magnesium; Soil K: soil potassium; NDF: neutro-detergent fibre; ADF: acid-detergent fibre; ADL: acid-detergent lignin;

NH: nitrogen in the herbage; CH: carbon in the herbage; CH/NH: carbon to nitrogen ratio in the

 herbage; Abiom: aboveground biomass; NDF/CP: neutro-detergent fibre to crude protein ratio;

 ADL/NH: acid-detergent lignin to nitrogen in the herbage ratio.

|  | Minimum | Maximum | Median | Mean |
|---|---|---|---|---|
| MAT | 1.08 | 9.90 | 4.72 | 4.96 |
| MST | 7.88 | 16.93 | 12.23 | 12.47 |
| TSIS | 6.80 | 7.80 | 7.58 | 7.51 |
| MAP | 964 | 1586 | 1252 | 1242.91 |
| MSP | 169.00 | 258.00 | 235.00 | 228.90 |
| Slope | 0.00 | 35.00 | 16.50 | 16.88 |
| Aspect | 1.00 | 3.00 | 1.84 | 2.05 |
| Sand | 3.10 | 72.20 | 32.80 | 32.67 |
| Loam | 13.60 | 73.50 | 38.60 | 39.80 |
| Clay | 2.90 | 68.60 | 27.25 | 27.53 |
| pH | 3.90 | 7.80 | 5.47 | 5.74 |
| Soil N | 0.11 | 1.10 | 0.46 | 0.47 |
| Soil P | 4.00 | 54.00 | 11.00 | 12.98 |
| Soil C/N | 4.13 | 41.60 | 12.47 | 13.39 |
| Soil Mg | 2.89 | 5.99 | 4.99 | 4.92 |
| Soil K | 3.40 | 6.84 | 4.99 | 5.03 |
| NDF | 31.20 | 78.90 | 52.45 | 52.08 |
| ADF | 17.70 | 46.60 | 29.55 | 30.07 |
| ADL | 1.16 | 12.72 | 6.32 | 6.63 |
| NH | 0.48 | 3.03 | 1.66 | 1.63 |
| CH | 22.60 | 49.10 | 45.15 | 44.53 |
| CH/NH | 13.90 | 97.20 | 26.60 | 31.14 |

| | | | | |
|---|---|---|---|---|
| Abiom | 64.52 | 1224 | 308.32 | 341.91 |
| NDF/CP | 2.15 | 17.20 | 4.77 | 5.71 |
| ADL/NH | 0.50 | 14.02 | 3.92 | 4.78 |

Table S3: Chemical composition of herbage samples used for NIRS calibration. DM: dry matter;

MM: mineral matter or ash content; CP: crude protein; NDF: neutro-detergent fibre; ADF: acid- detergent fibre; ADL: acid-detergent lignin; NH: nitrogen in the herbage; CH: carbon in the herbage.

| Parameter, % | N | Min. | Max. | Mean | SD |
|---|---|---|---|---|---|
| DM | 67 | 91.60 | 96.73 | 93.48 | 1.39 |
| MM (Ash) | 67 | 3.58 | 19.73 | 10.10 | 3.98 |
| CP | 67 | 5.50 | 14.67 | 9.29 | 1.90 |
| NDF | 67 | 36.82 | 73.11 | 55.42 | 9.27 |
| ADF | 67 | 21.95 | 41.97 | 30.00 | 4.70 |
| ADL | 67 | 3.35 | 12.52 | 6.18 | 2.08 |
| NH | 55 | 0.75 | 2.10 | 1.44 | 0.31 |
| CH | 55 | 36.83 | 51.13 | 45.10 | 2.99 |

Table S4: Calibration and cross validation statistics for predicting the chemical composition parameters in herbage samples by NIRS analysis. DM: dry matter; MM: mineral matter or ash content; CP: crude protein; NDF: neutro-detergent fibre; ADF: acid-detergent fibre; ADL: acid- detergent lignin; NH: nitrogen in the herbage; CH: carbon in the herbage.

| Parameter | Math[a] treatment | Scatter[b] correction | $R^2$ | $r^2$ | SEC | SECV | RPD |
|---|---|---|---|---|---|---|---|
| DM | 2,4,4,1 | DT | 0.92 | 0.85 | 0.392 | 0.539 | 2.58 |
| Ash | 2,4,4,1 | MSC | 0.83 | 0.70 | 1.583 | 0.830 | 4.80 |
| CP | 2,4,4,1 | SNV | 0.97 | 0.94 | 0.331 | 0.451 | 4.21 |
| NDF | 2,4,4,1 | DT | 0.83 | 0.72 | 3.756 | 4.728 | 1.96 |

| | | | | | | | |
|---|---|---|---|---|---|---|---|
| ADF | 2,4,4,1 | DT | 0.81 | 0.70 | 2.031 | 2.548 | 1.84 |
| ADL | 2,4,4,1 | MSC | 0.80 | 0.66 | 0.900 | 1.178 | 1.77 |
| N | 2,4,4,1 | MSC | 0.97 | 0.95 | 0.055 | 0.068 | 4.56 |
| C | 2,4,4,1 | MSC | 0.97 | 0.95 | 0.422 | 0.581 | 5.15 |

[a]Designations: derivate order, gap, first smoothing, and second smoothing; [b]Standard Normal

Variance (SNV), Detrend (DT) and Multiplicative Scattering Correction (MSC) transformations.

$R^2$ = coefficient of determination for calibration. $r^2$ = coefficient of determination for cross validation. SEC = standard error of calibration. SECV = standard error of cross validation. RPD = ratio of performance to deviation (RPD=SD/SECV).

Table S5: Variance inflation values for the continuous predictors included in the GLMs. Values under 5 are considered non-problematic (Heiberger, 2017). MAP: mean annual precipitation;

TSIS: mean summer temperature minus mean annual temperature; Slope: terrain slope; Clay:

clay content; Soil C/N: soil carbon to nitrogen ratio; soil N: soil nitrogen; NDF: neutro- detergent fibre; ADL/NH: acid-detergent lignin to nitrogen in the herbage ratio.

| Predictor | MAP | MMT | Slope | Clay | Log(soil C/N) | Soil N | NDF | ADL/NH |
|---|---|---|---|---|---|---|---|---|
| Geophysical model | 1.26 | 1.16 | 1.27 | 1.22 | - | - | - | - |
| Complete model | - | 1.26 | 1.32 | - | 1.58 | 1.82 | 1.32 | 1.67 |

[Figure]

Figure S1: Map of the study area. Points indicate sampling locations.

     Figure S2: Scheme of the sampling procedure for building the PASTUS database

[Figure]

[Figure]

[Figure]

[Figure]

**RFE deviance - SOC20 - folds = 100**

Figure S3: Changes in the predictive deviance of BRT models by backward removal of its predictors. The solid line indicates the mean change in predictive deviance, and the dotted line the standard error, calculated over the 10 folds of the cross-validation.  Solid vertical line indicates the variables removed for the second fit. Dotted vertical line indicates minimum change in predictive deviance. Dotted horizontal line indicates mean change in predictive deviance.

[Figure]

[Figure]

[Figure]

[Figure]

Figure S4: Histogram and normal Q-Q plot of A) SOC and B) log(SOC). Result of Shapiro Wilk W

test result were W = 0.948; p-value < 0.001 and W = 0.99; p-value = 0.18 respectively. SOC: soil organic carbon.

[Figure]

Figure S5: Linear modelling procedure. A) Variables introduced in each step. The first linear model (Geophysical model) is fitted until Step 2 and the second linear model (Complete

Model) is fitted until Step 4. B) For selecting the candidate predictor terms on each step, residuals of the model obtained in the previous step are used as response variables in C. C)

Procedure to select candidate terms on each step. First, genetic algorithm was used to obtain a set of best models. Second, these models were averaged and the significant terms were selected as candidates for backward forward selection in the main/consolidated model.

[Figure]

Figure S6: Relative contributions of variable groups in the linear model explaining Soil Organic

Carbon, using regional, landscape and management predictors.  MAP: mean annual precipitation; TSIS: mean summer temperature minus mean annual temperature; Slope:

terrain slope; Exposed: Exposed position according to Macrotopography; Clay: clay content;

Low and medium intensity: Low and medium intensity according to Grazing intensity.

[Figure]

Figure S7: Relative contributions of variable groups in the linear model explaining Soil Organic

Carbon using regional, landscape, management and biochemical predictors.  MAP: mean annual precipitation; TSIS: mean summer temperature minus mean annual temperature;

Slope: terrain slope; Cattle and Mixed: Cattle and mixed management according to grazing species; Low and medium intensity: Low and medium intensity according to Grazing intensity;

Soil C/N: soil carbon to nitrogen ratio; soil N: soil nitrogen; NDF: neutro-detergent fibre;

ADL/NH: acid-detergent lignin to nitrogen in the herbage ratio.

[Figure]

Figure S8: Partial dependence plots for the 15 selected predictors in the BRT model. Y axes are centred to have zero mean over data distribution. Values (solid lines) are predictions of the model across the predictor´s range maintaining the rest of the predictors at their average values. Grey areas around prediction lines indicate 95% bootstrap confidence intervals. Soil N:

soil nitrogen; Soil C/N: soil carbon to nitrogen ratio, Clay: clay content; Abiom: aboveground biomass; ADL: acid-detergent lignin; Silt: silt content; K: soil potassium; TSIS: mean summer temperature minus mean annual temperature; NDF: neutro-detergent fibre; pH: soil pH; CH:

carbon in the herbage; Mg: soil magnesium; Slope: terrain slope; MAP: mean annual precipitation; ADF: acid detergent fibre. See Table S1 for more details about variables.

[Figure]

Figure S9: Pairwise Pearson´s correlations between climate variables. MST: mean summer temperature; MWT: mean winter temperature; MAT: mean annual temperature; TSIS: inter- annual seasonality measured as MST-MAT.

---

## Author Response (AR2)

Answers to Editor:

Thanks a lot for carefully revising the manuscript. It reads much better now! I find it a great exploration of driving factors of SOC storage although I still find some sections rather speculative.

For instance at the first view, I have found TSIS a potentially great variable tackling seasonality but at a closer look the observed range seems close to the detection limit and thus somewhat ambiguous. Well, this is part of the discussion and identifying or presenting new controlling factors may stimulate other researchers to test them in their data set with a greater data span.

Looking at Figure 3, the range of TSIS seems very small (1°C, >70% of the values ranging less than 0.5°C), almost being in the range of measurement uncertainty, especially because TSIS is derived from two measurements and the relationship primarily be driven by three sites with low TSIS. Consequently, I find your conclusions that TSIS is a key driver somewhat too strong.

As we answered in the previous letter, we partially agree with you when you say some results are speculative. However, as you mentioned, one of the aims of our study is stimulate other researchers to test them, not only in observational but also in manipulative experiments. As we commented in the previous answers, there is an important lack in amount and variety of experiments testing interactions between drivers of soil properties (Rillig et al., 2019).

Focusing on your comments on TSIS:

We really appreciate your comments about TSIS, we find then really acute. Being conscious of the limitations of our analysis, we just refer TSIS as a "key driver of SOC" in **our study** (L 494). In the section before, we already explained that "it is not possible to unequivocally establish the causal links between SOC drivers" (L 460). Finally, the manuscript concludes that "we provided valuable information for further studies dealing with SOC predictions at broad several scales,

and laid out the basis to generate new testable hypotheses for future studies" (L 717). In our view, the kind of information that our study provides is clear enough.

You argued that TSIS was derived of two measurements, however, **TSIS is not derived from only two measurements, it is derived from two variables**. TSIS is the difference between the mean summer temperature and mean annual temperature. Attending to the documentation of Worldclim 2.0 (Fick and Hijmans, 2017), each one of these variables were obtained from the annual means of the corresponding months, using data from weather stations which **comprised a 30 year period** (1970–2000). Additionally, note that temperature variables from WorldClim 2.0 had a very high accuracy. All temperature variables had a global correlation coefficient (between estimated and observed values) of 0.99. **Therefore, if one temperature variable of WorldClim 2.0 increases, the actual value surely increases in a similar way.** Of course climatic models like WorldClim 2.0 have their limitations, but considering the years and the amount of weather stations they compile for the modelling, they are nowadays the best possible available source of climate information considering the sample size and extent of our study.

We would like to compare TSIS with the temperature seasonality index, which can be found in many papers using bioclimatic variables a predictors (i.e. see Rodríguez et al., 2015; Cano et al.. 2018 and the references therein). Temperature seasonality index is the standard deviation of the mean temperature of each of the 12 months. This is, it gives information of how far are the mean temperatures of the 12 months from the mean. What we do with TSIS is considering just the difference between the warmest months (that is, the summer months) and the mean. Note in the following maps that both indexes look similar, but they have their differences, since TSIS just represents summer variations instead of annual variations. **TSIS has more ecological sense for us, since summer is the period when the productivity peak of mountain grasslands occurs** (Gómez, 2008). In addition, when we used SOC stocks models including the two variables, TSIS

and the temperature seasonality index, TSIS always compared favourably, suggesting higher sensitivity to ecological variables than the other seasonality index.

Interestingly, TSIS has provided good results in preliminary SOC modelling for our analysis and also in other works by the group, using other variables in the same (species richness, Rodríguez et al., 2018) or a different database (soil activity variables; Debouk et al., 2020).

[Figure]

Figure: TSIS and standard deviation of the annual mean temperature (°C) according to WorldClim 2.0 in the Pyrenees (altitude > 1200 m).

Finally, we did a little exercise to determinate to what extent the relationship between TSIS and SOC stocks in our model is primarily driven by three sites with low TSIS, as you commented.

Firstly, we re-fit the geophysical model without the 3 sites with the lowest TSIS. Looking to the significance of the model terms (compare the table below with Table 2), although there was some variation in some estimates, **both the significance levels and the sing of the effects remained the same as in Table 2.**

Table: Results of the geophysical model for soil organic carbon excluding the 3 cases with the lowest TSIS values ($R^2_{Adj}$ = 0.3). Compare with Table 3.

| Model term | Estimate | SE | t-value | P-value | |
|---|---|---|---|---|---|
| Intercept | 1.18 | 2.10 | 0.56 | 0.57 | |
| **Climate variables** | | | | | |
| MAP | 0.003 | 0.001 | 4.79 | <0.001 | *** |
| TSIS | -0.35 | 0.26 | -1.32 | 0.19 | |
| **Topography variables** | | | | | |
| Slope | -0.38 | 0.12 | -3.32 | 0.001 | ** |
| Exposed | -2.01 | 1.01 | -1.99 | 0.049 | * |
| **Soil type variables** | | | | | |
| Clay | 0.11 | 0.03 | 4.24 | < 0.001 | *** |
| **Management variables** | | | | | |
| Low intensity | -3.29 | 1.58 | -2.09 | 0.04 | * |
| Medium intensity | 2.02 | 1.27 | 1.59 | 0.11 | |
| **Interactions** | | | | | |
| MAPxClay | $9*10^{-5}$ | $3*10^{-5}$ | -4.37 | <0.001 | *** |
| TSISxExposed | 0.27 | 0.13 | 2.02 | 0.05 | |
| TSISxLow intensity | 0.43 | 0.21 | 2.07 | 0.046 | * |
| TSISxMedium intensity | -0.27 | 0.17 | -1.57 | 0.12 | |
| TSIS:Slope | 0.05 | 0.02 | 3.24 | 0.002 | ** |

To illustrate this, we draw the plots of the TSISxMacrotopography and TSISxGrazing intensity effects, in the same way as we did in the manuscript (compare the plot below with Fig. 3). Although the effects are weaker, **we can appreciate the same effects.**

[Figure]

Figure: TSIS and macrotopography and TSIS and grazing intensity effects on SOC according to the model fitted excluding the 3 cases with the lowest TSIS values. Compare with Fig. 3.

Secondly, we refit the geophysical model using the function "lmrobust" in the "robustbase" R package. This function fits linear models with robust estimates (Yaffee, 2002). In a few words, models are built with conservative estimates that take less into account the cases which are the most extreme outliers.

Comparing the table below with Table 3, **we can appreciate that the sign and the significance of the geophysical model still the same if we use robust estimates instead of conventional ones.**

Table: Results of the geophysical model for soil organic carbon excluding the 3 cases with the lowest TSIS values ($R^2_{Adj}$ = 0.38). Compare with Table 3.

| Model term | Estimate | SE | t-value | P-value |
|---|---|---|---|---|
| Intercept | -0.19 | 1.84 | -0.10 | 0.92 |
| **Climate variables** | | | | |
| MAP | 0.003 | 0.001 | 4.48 | 0.00 |
| TSIS | -0.14 | 0.23 | -0.60 | 0.55 |
| **Topography variables** | | | | |
| Slope | -0.36 | 0.10 | -3.67 | 0.00 |
| Exposed | -3.10 | 0.96 | -3.24 | 0.00 |
| **Soil type variables** | | | | |
| Clay | 0.12 | 0.03 | 4.28 | 0.00 |
| **Management variables** | | | | |
| Low intensity | -5.18 | 1.22 | -4.25 | 0.00 |
| Medium intensity | 2.21 | 1.20 | 1.83 | 0.07 |
| **Interactions** | | | | |
| MAPxClay | $9*10^{-5}$ | $3*10^{-5}$ | -4.44 | 0.00 |
| TSISxExposed | 0.41 | 0.13 | 3.25 | 0.00 |
| TSISxLow intensity | 0.68 | 0.16 | 4.17 | 0.00 |
| TSISxMedium intensity | -0.29 | 0.16 | -1.81 | 0.07 |
| TSISxSlope | 0.05 | 0.01 | 3.55 | 0.00 |

We think that these explanations support properly our interpretation of the results: TSIS was an important factor that drove SOC in interaction with other variables. The 3 lowest TSIS values provide valuable information to the model, but they are not necessary to get similar results in significance and sign. In other words, the relationship is not primarily driven by those three sites. Moreover, even with an apparently low range we get robust and significant effects for this variable.

To conclude: TSIS has a low range of variation, but ecological variables (including SOCS stocks in this study; plant species richness in Rodriguez et al. 2018; soil activity variables in Debouk et al.

2020) show high sensitivity within this range; we expect other authors to test this variable further, and our own group is considering testing TSIS further; TSIS performed favourably compared to similar seasonality indices previously used, showing higher sensitivity to ecological variables than those.

In a similar sense at

L. 554 'contrast with most other' – this is not correct, the other studies have simply not included TSIS in their set of parameters – I would rather argue that your data suggest to include it in other studies which may also test it for stronger gradients than in this study.

We think this is a judicious suggestion, we changed the text as follows:

"While most of the previous studies addressing soil carbon not included any temperature seasonality variable as potential SOC predictor, usually focusing in mean temperature and precipitation as the most important climate drivers of SOC (Hobley et al., 2015; Manning et al., 2015; Wiesmeier et al., 2019), our models suggest that TSIS and other temperature seasonality indexes should be included in further studies, to provide more evidence of the extent of the effects of temperature seasonality on SOC stocks." (L 544).

Please carefully check the use of upper and lower case throughout the whole manuscript (see examples below).

We checked the use of upper and lower case throughout the whole manuscript.

426 change 88.31% to 88%

Change done.

L. 427 change Clay to clay

Change done.

L. 435 Combined

Change done.

L. 436 add empty space before (Fig…)

Change done.

L. 439 Geophysical Model

Change done.

L. 503 the sentence reads incorrect, especially the last half.

We changed the phrase: "This increase in soil organic matter inputs during summer would overcome an eventual increase of soil organic matter decomposition caused by high temperatures (Sanderman et al., 2003)." (L 503)

L. 561 Cold Sites

Change done.

**References**

Cano, J., Rodríguez, A., Simpson, H., Tabah, E. N., Gómez, J. F. and Pullan, R. L.: Modelling the spatial distribution of aquatic insects (Order Hemiptera) potentially involved in the transmission of Mycobacterium ulcerans in Africa, Parasites and Vectors, 11(1), doi:10.1186/s13071-018-3066-3, 2018.

Debouk, H., San Emeterio, L., Marí, T., Canals, R. M. and Sebastià, M.-T.: Plant Functional Diversity, Climate and Grazer Type Regulate Soil Activity in Natural Grasslands, Agronomy, 10(9), 1291, doi:10.3390/agronomy10091291, 2020.

Fick, S. E. and Hijmans, R. J.: WorldClim 2: new 1-km spatial resolution climate surfaces for global land areas, Int. J. Climatol., 37(12), 4302–4315, doi:10.1002/joc.5086, 2017.

Gómez, D.: Aspectos ecológicos de los pastos, in Pastos del Pirineo, edited by F. Fillat, R. García-González, D. Gómez, and R. Reiné, pp. 61–73, Consejo Superior de Investigaciones Científicas, CSIC, Madrid, Spain., 2008.

Rillig, M. C., Ryo, M., Lehmann, A., Aguilar-Trigueros, C. A., Buchert, S., Wulf, A., Iwasaki, A., Roy, J. and Yang, G.: The role of multiple global change factors in driving soil functions and microbial biodiversity, Science (80-. )., 366(6467), 886–890, doi:10.1126/science.aay2832, 2019.

Rodríguez, A., Gómez, J. F. and Nieves-Aldrey, J. L.: Modeling the potential distribution and conservation status of three species of oak gall wasps (Hymenoptera: Cynipidae) in the Iberian range, J. Insect Conserv., 19(5), doi:10.1007/s10841-015-9810-5, 2015.

Rodríguez, A., de Lamo, X., Sebastiá, M.-T.: Interactions between global change components drive plant species richness patterns within communities in mountain grasslands independetly of topography, edited by B. Collins, J. Veg. Sci., 29(August), 1029–1039, doi:10.1111/jvs.12683, 2018.

Yaffee, R. A.: Robust regression analysis: Some popular statistical package options, ITS Stat.

Soc. Sci. Mapp. Group, New York State Univ. downloaded Dec, 23, 156–162, 2002.